# Initiator tRNA lacking 1-methyladenosine is targeted by the rapid tRNA decay pathway in evolutionarily distant yeast species

**Monika Tasak**[ORCID], **Eric M. Phizicky**[ORCID]*

Department of Biochemistry and Biophysics, Center for RNA Biology, University of Rochester School of Medicine, Rochester, New York, United States of America

* eric_phizicky@urmc.rochester.edu

## Abstract

All tRNAs have numerous modifications, lack of which often results in growth defects in the budding yeast *Saccharomyces cerevisiae* and neurological or other disorders in humans. In *S. cerevisiae*, lack of tRNA body modifications can lead to impaired tRNA stability and decay of a subset of the hypomodified tRNAs. Mutants lacking 7-methylguanosine at $G_{46}$ ($m^7G_{46}$), $N_2$,$N_2$-dimethylguanosine ($m^{2,2}G_{26}$), or 4-acetylcytidine ($ac^4C_{12}$), in combination with other body modification mutants, target certain mature hypomodified tRNAs to the rapid tRNA decay (RTD) pathway, catalyzed by 5'-3' exonucleases Xrn1 and Rat1, and regulated by Met22. The RTD pathway is conserved in the phylogenetically distant fission yeast *Schizosaccharomyces pombe* for mutants lacking $m^7G_{46}$. In contrast, *S. cerevisiae trm6/gcd10* mutants with reduced 1-methyladenosine ($m^1A_{58}$) specifically target pre-tRNA$_i$^Met(CAU) to the nuclear surveillance pathway for 3'-5' exonucleolytic decay by the TRAMP complex and nuclear exosome. We show here that the RTD pathway has an unexpected major role in the biology of $m^1A_{58}$ and tRNA$_i$^Met(CAU) in both *S. pombe* and *S. cerevisiae*. We find that *S. pombe trm6Δ* mutants lacking $m^1A_{58}$ are temperature sensitive due to decay of tRNA$_i$^Met(CAU) by the RTD pathway. Thus, *trm6Δ* mutants had reduced levels of tRNA$_i$^Met(CAU) and not of eight other tested tRNAs, overexpression of tRNA$_i$^Met(CAU) restored growth, and spontaneous suppressors that restored tRNA$_i$^Met(CAU) levels had mutations in *dhp1/RAT1* or *tol1/MET22*. In addition, deletion of *cid14/TRF4* in the nuclear surveillance pathway did not restore growth. Furthermore, re-examination of *S. cerevisiae trm6* mutants revealed a major role of the RTD pathway in maintaining tRNA$_i$^Met(CAU) levels, in addition to the known role of the nuclear surveillance pathway. These findings provide evidence for the importance of $m^1A_{58}$ in the biology of tRNA$_i$^Met(CAU) throughout eukaryotes, and fuel speculation that the RTD pathway has a major role in quality control of body modification mutants throughout fungi and other eukaryotes.

**Data Availability Statement:** All relevant data are within the manuscript and its supporting information files.

**Funding:** This research was supported by Grant GM052347, awarded to EMP from the National Institute of General Medical Sciences of the National Institutes of Health (https://www.nigms.nih.gov/). MT was partially supported by NIH Training Grant T32 GM068411 in Cellular, Biochemical and Molecular Sciences. The funders had no role in study design, data collection and analysis, decision to publish, or preparation of the manuscript.

**Competing interests:** The authors have declared that no competing interests exist.

## Author summary

tRNA modifications are highly conserved, and their lack frequently results in growth defects in the yeast *Saccharomyces cerevisiae* and neurological disorders in humans. In *S. cerevisiae* lack of 1-methyladenosine at $N_{58}$ ($m^1A_{58}$) in the tRNA body is lethal due to 3'-5' decay of pre-tRNA$_i^{Met}$ by the nuclear surveillance pathway. By contrast, lack of any of three other body modifications causes growth defects due to 5'-3' decay of specific hypo-modified tRNAs by the rapid tRNA decay (RTD) pathway. Despite their importance, little is known about either tRNA$_i^{Met}$ quality control or tRNA decay pathways in eukaryotes other than *S. cerevisiae*.

Here we show an unexpected role of the RTD pathway in quality control of tRNA$_i^{Met}$ lacking $m^1A_{58}$ in the phylogenetically distant yeast species *Schizosaccharomyces pombe* and *S. cerevisiae*. We find that *S. pombe trm6Δ* mutants, lacking $m^1A_{58}$, are temperature sensitive due to decay of tRNA$_i^{Met(CAU)}$ primarily by the RTD pathway. Furthermore, re-investigation of *S. cerevisiae trm6* mutants revealed a significant role of the RTD pathway, in addition to the nuclear surveillance pathway, in decay of tRNA$_i^{Met(CAU)}$. Our results suggest that throughout eukaryotes the RTD pathway has a major role in decay of hypo-modified tRNAs and that $m^1A_{58}$ is crucial to tRNA$_i^{Met(CAU)}$ biology.

## Introduction

tRNAs are central to the process of translation, a role that is enabled by their extensive and highly conserved post-transcriptional modifications [1–3]. Lack of any of a number of modifications causes growth defects in the budding yeast *Saccharomyces cerevisiae* [3], as well as a number of neurological or mitochondrial disorders in humans [4, 5]. Lack of modifications in and around the anticodon loop (ACL) frequently reduces the efficiency and/or fidelity of mRNA decoding [6–8], disrupts reading-frame maintenance [9, 10], or decreases charging efficiency and/or fidelity [11]. Lack of modifications in the main tRNA body (outside the ACL) often results in altered folding [12] or reduced tRNA stability, leading to degradation of a subset of the hypomodified tRNAs by one of two characterized decay pathways [13–15].

The rapid tRNA decay (RTD) pathway degrades a subset of the tRNA species lacking any of several body modifications. Degradation of tRNAs by the RTD pathway is catalyzed by the 5'-3' exonucleases Rat1 and Xrn1, and inhibited by a *met22Δ* mutation [16] due to accumulation of the Met22 substrate adenosine 3',5' bisphosphate (pAp) [17, 18]. In *S. cerevisiae*, lack of $m^7G_{46}$, $m^{2,2}G_{26}$, or $ac^4C_{12}$ is known to trigger RTD, and is associated with temperature sensitivity, particularly in combination with lack of other tRNA body modifications. Thus, an *S. cerevisiae trm8Δ trm4Δ* mutant (lacking $m^7G_{46}$ and $m^5C$), is temperature sensitive due to decay of mature tRNA$^{Val(AAC)}$ by the RTD pathway [14, 16]. Similarly, an *S. cerevisiae tan1Δ trm44Δ* mutant (lacking $ac^4C_{12}$ and $Um_{44}$) and a *trm1Δ trm4Δ* mutant (lacking $m^{2,2}G_{26}$ and $m^5C$) are each temperature sensitive due to decay of tRNA$^{Ser(CGA)}$ and tRNA$^{Ser(UGA)}$ [16, 19, 20]. Moreover, each of the corresponding *trm8Δ*, *tan1Δ*, and *trm1Δ* single mutants has an RTD signature, as their temperature sensitivity is suppressed by a *met22Δ* mutation, and is associated with decay of one or more hypomodified tRNA substrates [20].

In addition, recent results show that the RTD pathway also acts on a subset of tRNA species lacking $m^7G_{46}$ in the phylogenetically distant fission yeast *Schizosaccharomyces pombe*. Thus, the temperature sensitivity of *S. pombe trm8Δ* mutants is due to decay of tRNA$^{Tyr(GUA)}$ and to some extent tRNA$^{Pro(AGG)}$, and both the decay and the temperature sensitivity are suppressed by mutations in the *RAT1* ortholog *dhp1* [15].

The other major decay pathway that targets tRNAs lacking a body modification in *S. cerevisiae* is the nuclear surveillance pathway, which degrades the precursor of initiator tRNA (pre-tRNA$_i^{Met(CAU)}$) lacking m$^1$A$_{58}$ [13, 21]. Degradation by this pathway is catalyzed by Trf4 of the TRAMP complex, which oligoadenylates the 3' end of pre-tRNA$_i^{Met(CAU)}$, followed by its 3'-5' exonucleolytic degradation by Rrp6 and Rrp44 of the nuclear exosome [13, 22–25]. The m$^1$A$_{58}$ modification is found on numerous tRNA species in *S. cerevisiae*, but tRNA$_i^{Met(CAU)}$ is uniquely different from other tRNAs due in part to its non-canonical nucleotides at A$_{20}$, A$_{54}$, and A$_{60}$ and an unusual substructure involving these residues and m$^1$A$_{58}$ [26], presumably accounting for its unique sensitivity to decay in strains lacking m$^1$A$_{58}$ [13]. In addition, the nuclear surveillance pathway is also known to target about 50% of all tRNA transcripts, possibly due to stochastic errors during transcription, mis-folding, or natural competition between decay and processing [27].

Understanding the quality control of tRNA$_i^{Met(CAU)}$ is crucial because of its central role in translation initiation. In this role, tRNA$_i^{Met(CAU)}$ is a component of the eukaryotic ternary complex that binds the 40S ribosome subunit to form the 43S pre-initiation complex, which in turn binds capped mRNAs and scans their sequence for an appropriate AUG start codon [28]. Moreover, tRNA$_i^{Met(CAU)}$ is only involved in translation initiation and has its own dedicated set of factors for its delivery to the 40S subunit [29], whereas all other tRNAs participate only in elongation, and their delivery only involves the elongation factor eEF-1A [30], which does not participate in translation initiation. In addition, tRNA$_i^{Met(CAU)}$ levels are important in regulating the general amino acid control pathway (integrated stress response in humans) by regulation of translation of the transcription factor Gcn4 (human ATF4), which re-programs gene expression in the cell [31, 32]. Furthermore, in human breast epithelial cells, overexpression of tRNA$_i^{Met(CAU)}$ causes increased cell proliferation and metabolic activity [33], and in mouse, elevated expression of tRNA$_i^{Met(CAU)}$ stimulates cell migration and drives melanoma invasion [34].

Despite the importance of tRNA$_i^{Met(CAU)}$ levels in *S. cerevisiae* and humans, it is not clear how tRNA$_i^{Met(CAU)}$ levels are regulated in eukaryotes. Although there is compelling evidence in *S. cerevisiae* that the nuclear surveillance pathway targets pre-tRNA$_i^{Met(CAU)}$ for decay in *trm6/gcd10* or *trm61/gcd14* mutants with reduced m$^1$A$_{58}$, available information in other eukaryotes suggests the possibility of alternative pathways. HeLa cells that are heat shocked at 43°C undergo decay of tRNA$_i^{Met(CAU)}$ by Xrn1 and Rat1 over several hours, but it is not clear why decay occurred, as there was no obvious change in the modifications or physical stability of the remaining tRNA$_i^{Met(CAU)}$ [35]. Moreover, tRNA$_i^{Met(CAU)}$ levels in human cells were upregulated by knockdown of ALKBH1, which has an m$^1$A-demethylase activity, and glucose starvation led to increased ALKBH1, linked to reduced tRNA$_i^{Met(CAU)}$ and reduced translation [36]. Although the mechanisms by which tRNA$_i^{Met(CAU)}$ levels are regulated are not known, it is clear that there is a link between m$^1$A$_{58}$ modification status and tRNA$_i^{Met(CAU)}$ levels in eukaryotes. Thus, depletion of TRM6 or TRM61 in human cells results in reduced levels of tRNA$_i^{Met(CAU)}$ and slow growth, which is partially rescued by overexpression of tRNA$_i^{Met(CAU)}$ [37]. Similarly, *Arabidopsis thaliana trm61* mutants have reduced tRNA$_i^{Met(CAU)}$ levels, which is associated with shortened siliques [38].

To address the evolutionary role and mechanisms by which m$^1$A$_{58}$ influences tRNA$_i^{Met(CAU)}$ levels and function, we have compared m$^1$A$_{58}$ biology in the fission yeast *S. pombe* with that in *S. cerevisiae*, which diverged ~600 million years ago (Mya) [39]. We show here that the RTD pathway has an unexpected major role in the biology of m$^1$A$_{58}$ and tRNA$_i^{Met(CAU)}$ in both *S. pombe* and *S. cerevisiae*. We find that *S. pombe trm6Δ* mutants lack m$^1$A$_{58}$ and are temperature sensitive due to the decay of tRNA$_i^{Met(CAU)}$ by the RTD pathway, as spontaneous suppressors that restored tRNA$_i^{Met(CAU)}$ levels had mutations in the *RAT1* ortholog *dhp1* or the *MET22*

ortholog *tol1*, whereas mutation of the *TRF4* ortholog *cid14* did not suppress the growth defect. Moreover, we found a major role of the RTD pathway in *S. cerevisiae TRM6* biology, as mutation of the RTD pathway components *MET22*, *RAT1*, or *XRN1* each suppressed the temperature sensitivity of *S. cerevisiae trm6-504* mutants and restored tRNA$_i$$^{Met(CAU)}$ levels. Furthermore, we found that the lethality of *S. cerevisiae trm6Δ* mutants was suppressed by inhibition of both the RTD pathway and the nuclear surveillance pathway, but not by inhibition of either pathway alone.

Thus, our results show the importance of tRNA$_i$$^{Met(CAU)}$ as a target for m$^1$A$_{58}$ modification by Trm6:Trm61 across evolutionarily distant fungal species. Our results also uncover an unexpected conserved evolutionary role of the RTD pathway in tRNA$_i$$^{Met(CAU)}$ quality control in *trm6* mutants of both fungal species, as previously found for all three other body modification mutants studied in *S. cerevisiae* [16, 20], and the only other body modification mutant studied in *S. pombe* [15]. These findings fuel speculation that the RTD pathway has a major role in quality control of other body modification mutants in these organisms, as well as in metazoans.

## Results

### *S. pombe trm6Δ* and *trm61Δ* mutants are temperature sensitive, and lack m$^1$A$_{58}$ in their tRNA

To begin analysis of the biology of *S. pombe* Trm6 and Trm61, we investigated the growth phenotype of *trm6Δ* and *trm61Δ* mutants which, unlike the corresponding *S. cerevisiae* mutants, are reported to be viable [40]. To guard against background mutations that might have accumulated in the deletion collection, we first re-made the *trm6Δ* and *trm61Δ* mutants in a wild type (WT) strain, using appropriate kanamycin resistance cassettes, and then compared the growth of two independent *trm6Δ* and *trm61Δ* transformants relative to the WT parent. Each tested *S. pombe trm6Δ* and *trm61Δ* mutant grew nearly as well as WT at lower temperatures, but was temperature sensitive on rich (YES) media at 38˚C, and on minimal complete media lacking histidine (EMMC-His) at 33˚C (Fig 1A). As expected if these *trm6Δ* and *trm61Δ* phenotypes were due to the corresponding deletions, the growth defects were fully complemented after introduction of an *S. pombe* [*leu2*$^+$] plasmid expressing P$_{trm6}$ *trm6*$^+$ or P$_{trm61}$ *trm61*$^+$ respectively (S1A and S1B Fig).

Because the Trm6:Trm61 complex is essential in *S. cerevisiae* for m$^1$A$_{58}$ modification of substrate tRNAs [21, 41], we examined *S. pombe trm6Δ* and *trm61Δ* mutants for m$^1$A$_{58}$, to guard against the possibility that the mutants were alive because there is another protein that can catalyze some m$^1$A$_{58}$ modification. Examination by HPLC of the nucleoside composition of purified tRNA$^{Tyr(GUA)}$ from *trm6Δ* and *trm61Δ* mutants revealed that m$^1$A levels were less than 0.03 moles/mole, compared to 0.60 moles/mole in WT cells, whereas levels of Ψ, m$^5$C, and m$^7$G were very similar in the tRNA$^{Tyr(GUA)}$ from both mutant and WT cells (Fig 1B and 1C). Poison primer extension of tRNA$^{Tyr(GUA)}$ from WT bulk RNA showed a complete block at U$_{59}$ (98%) due to the presence of m$^1$A$_{58}$, which was virtually undetectable in *trm6Δ* and *trm61Δ* mutants (0.1% for *trm6Δ* and 0.8% in *trm61Δ*) (Fig 1D and 1E). Similarly, poison primer extension showed that tRNA$_i$$^{Met(CAU)}$ was nearly completely modified with m$^1$A$_{58}$ in WT cells (97%), but not visibly modified in *trm6Δ* and *trm61Δ* mutants (although quantification with the high background gave 2.0% for *trm6Δ* and 2.8% in *trm61Δ*). Furthermore, analysis of bulk tRNA modifications revealed that m$^1$A modification was less than 0.03% of the levels of cytidine in the *trm6Δ* mutant, compared to 2.5% for WT, whereas levels of Ψ were similar in both strains (16.8% vs 17.9%), as were levels of m$^5$C, m$^2$G, m$^7$G, and inosine (I) (S2

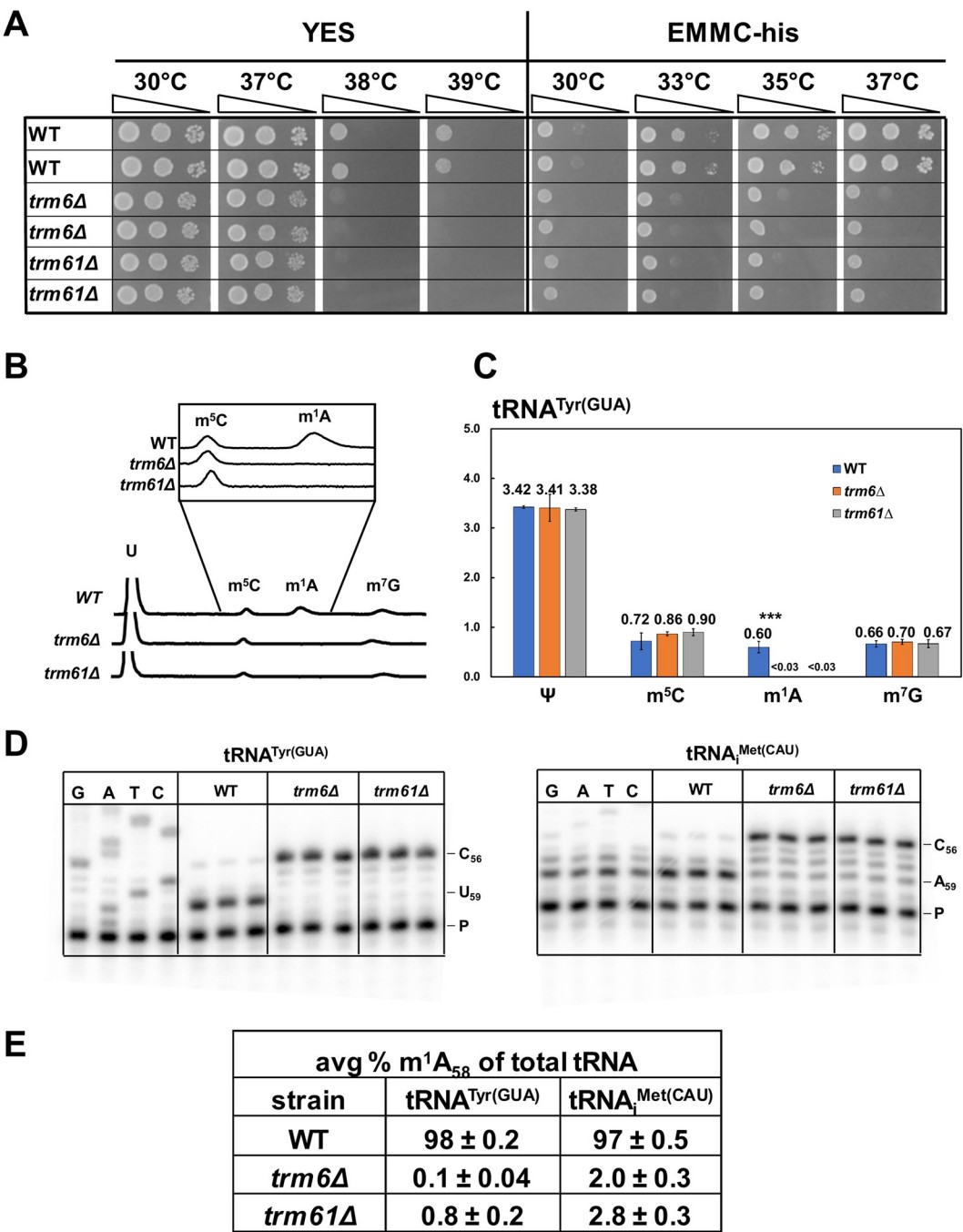

**Fig 1. *S. pombe trm6Δ* mutants and *trm61Δ* mutants are temperature sensitive and lack m$^1$A$_{58}$. (A) *S. pombe trm6Δ* and *trm61Δ* mutants are temperature sensitive on YES and EMMC-his media.** *S. pombe trm6Δ* mutants, *trm61Δ* mutants, and WT cells were grown overnight in YES media at 30°C, diluted to OD$_{600}$ ~0.5, serially diluted 10-fold in YES media, and then 2 μL were spotted onto plates containing YES or EMMC-his media and incubated at the indicated temperatures for 3 days. **(B) tRNA$^{Tyr(GUA)}$ from *S. pombe trm6Δ* and *trm61Δ* mutants has no detectable m$^1$A, as measured by HPLC separation of nucleosides.** *S. pombe trm6Δ* mutants, *trm61Δ* mutants, and WT cells were grown in biological triplicate in YES media at 30°C and tRNA$^{Tyr(GUA)}$ was purified, and digested to nucleosides, and then nucleosides were separated by HPLC as described in Materials and Methods. **(C) Quantification of levels of modified nucleosides of purified tRNA$^{Tyr(GUA)}$ in *S. pombe trm6Δ*, *trm61Δ*, and WT strains.** The chart shows average moles/mol of nucleosides with associated standard deviations; WT, blue; *trm6Δ*, orange; *trm61Δ*, gray. **(D) tRNA$^{Tyr(GUA)}$ and tRNA$_i$$^{Met(CAU)}$ from *S. pombe trm6Δ* and *trm61Δ* mutants have little or no detectable m$^1$A$_{58}$.** Bulk RNA from the growth for Fig 1B was analyzed by poison primer extension assay, as described in Materials and Methods, with primer OMT 775 (complementary to tRNA$_i$$^{Met(CAU)}$ nt 76–61) and primer OMT 477 (complementary to tRNA$^{Tyr(GUA)}$ 76–61) in the presence of ddGTP. The poison primer extension

produces a stop at $C_{56}$ for both tRNA$_i^{Met(CAU)}$ and tRNA$^{Tyr(GUA)}$, and the presence of m$^1$A$_{58}$ results in a stop at $N_{59}$. A sequencing ladder is shown at the left. *(E)* **Quantification of poison primer extension of tRNA$^{Tyr(GUA)}$ and tRNA$_i^{Met(CAU)}$.** For each primer extension, the signals at $N_{59}$ and $C_{56}$ were first corrected by subtraction of the signals at $A_{58}$ and $N_{57}$ respectively.

Fig). These results show that *S. pombe trm6$^+$* and *trm61$^+$* are required for all detectable m$^1$A$_{58}$ modification of cytoplasmic tRNAs.

## *S. pombe trm6Δ* mutants are temperature sensitive due to reduced levels of tRNA$_i^{Met(CAU)}$

Since the temperature sensitive growth defect of *S. cerevisiae trm6-504* mutants is caused by decreased levels of tRNA$_i^{Met(CAU)}$ [21], we analyzed levels of tRNA$_i^{Met(CAU)}$ and other tRNAs in *S. pombe trm6Δ* mutants at low and high temperatures to determine if this property was conserved. We sampled cells grown in rich media in triplicate at 30˚C and at 3-hour intervals after shift to 38.5˚C, and analyzed tRNA levels by northern blot hybridization. We quantified tRNA levels by normalizing relative to tRNA$^{Gly(GCC)}$ at the corresponding temperature and time point, and then relative to the normalized amount in WT cells at 30˚C. Note that, as shown previously [15], levels of the usual standards 5S and 5.8S rRNA are significantly affected by temperature changes in *S. pombe*, and tRNA$^{Gly(GCC)}$ levels were not affected in WT cells. Note also that in this and all other temperature shift experiments described in this report, cells were shown quantitatively to survive the temperature shift, based on spot tests of the cells on plates.

The northern analysis revealed that tRNA$_i^{Met(CAU)}$ levels were substantially reduced in the *S. pombe trm6Δ* mutants, both at 30˚C and 38.5˚C. At 30˚C, tRNA$_i^{Met(CAU)}$ levels were 49% of those in WT cells, whereas each of the other eight tRNAs had levels between 82% and 121% of those in WT cells (Figs 2A, 2B and S3). At 38.5˚C, tRNA$_i^{Met(CAU)}$ levels decreased further in the *trm6Δ* mutants, to 30% of WT levels after 3 hours, whereas levels of each of the other 8 tRNAs ranged from 78% to 114% of WT levels. Furthermore, although tRNA levels generally decreased after longer times at 38.5˚C, tRNA$_i^{Met(CAU)}$ levels decreased the most, to 19% of those in WT after 9 hours, compared to 53% to 90% for the other eight tRNAs. These results indicate that *S. pombe trm6Δ* mutants are associated with reduced levels of tRNA$_i^{Met(CAU)}$ at 30˚C, and with further reduced tRNA$_i^{Met(CAU)}$ levels at 38.5˚C.

To test if the temperature sensitivity of *S. pombe trm6Δ* mutants is caused by decreased levels of tRNA$_i^{Met(CAU)}$, we examined suppression of the *trm6Δ* growth defect upon tRNA$_i^{Met(CAU)}$ overexpression. In *S. pombe*, only one of the four genes encoding tRNA$_i^{Met(CAU)}$ is expressed in the usual manner, as a stand-alone tRNA gene. By contrast, the other three tRNA$_i^{Met(CAU)}$ genes are expressed in tandem with a tRNA$^{Ser}$ species as a tRNA$^{Ser}$-tRNA$_i^{Met}$ dimeric transcript, which is then processed into single tRNAs [42], similar to the tRNA$^{Arg}$-tRNA$^{Asp}$ tandem genes in *S. cerevisiae* [43, 44]. To test for suppression of the *trm6Δ* growth defect, we overexpressed either the stand-alone *SPBTRNAMET.06 (imt06$^+$)* gene, or the tandem tRNA$^{Ser}$-tRNA$_i^{Met}$ gene pair with *sup9$^+$* and *SPCTRNAMET.07 (imt07$^+$)*, each on a *[leu2$^+$]* plasmid. We found that the temperature sensitive growth defect of *S. pombe trm6Δ* mutants on EMMC-leu media was completely suppressed by expression of the stand-alone *imt06$^+$* gene, growing identically to that of an *S. pombe trm6Δ* [P$_{trm6}$ *trm6$^+$*] strain at high temperature (Fig 2C), whereas expression of *imt07$^+$* from the tRNA$^{Ser}$-tRNA$_i^{Met}$ tandem gene only modestly suppressed the *trm6Δ* temperature sensitivity. Consistent with this result, northern analysis showed that levels of tRNA$_i^{Met(CAU)}$ are much higher in strains overexpressing *imt06$^+$* than in strains overexpressing *imt07$^+$* (S4 Fig). Similarly, the temperature sensitivity of the *S. pombe trm61Δ*

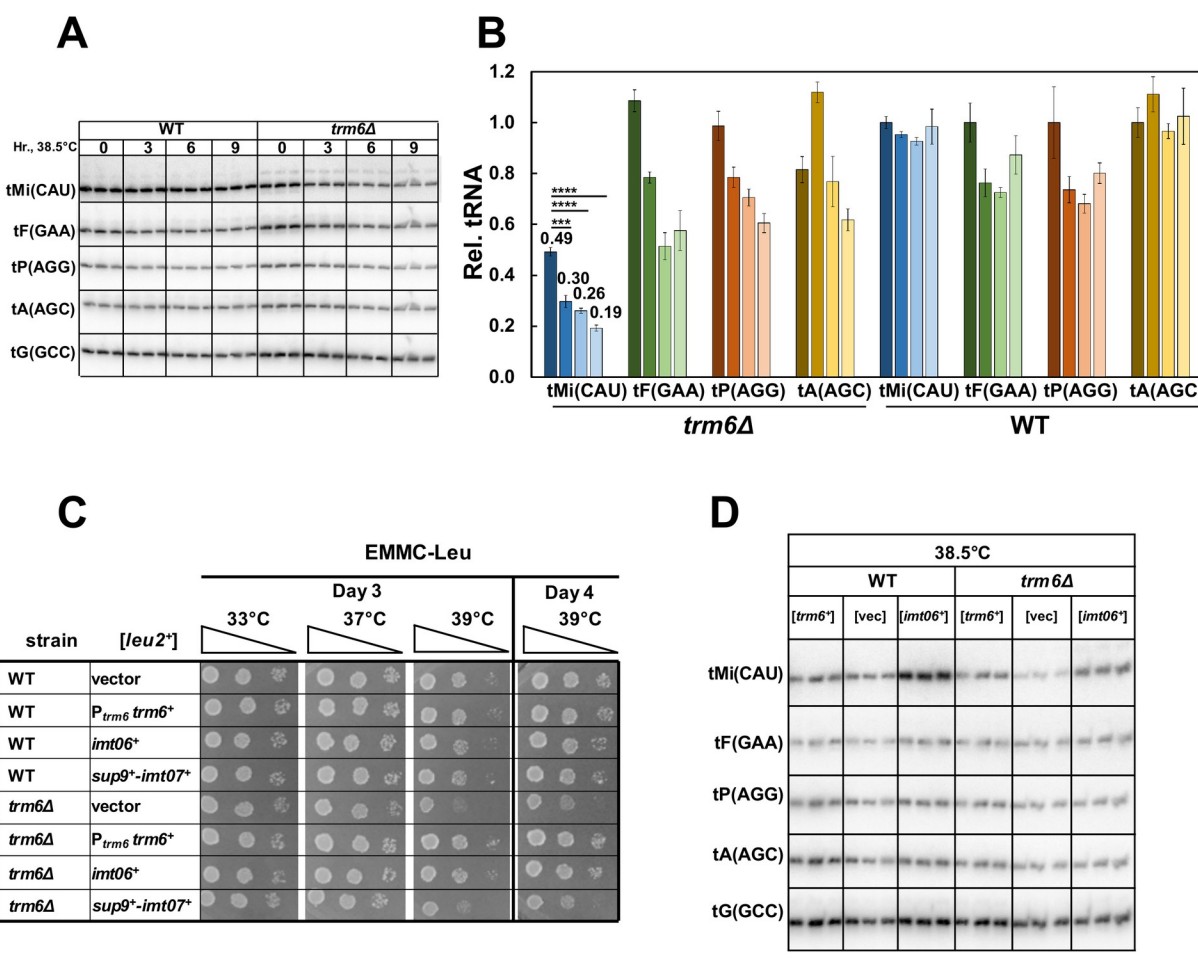

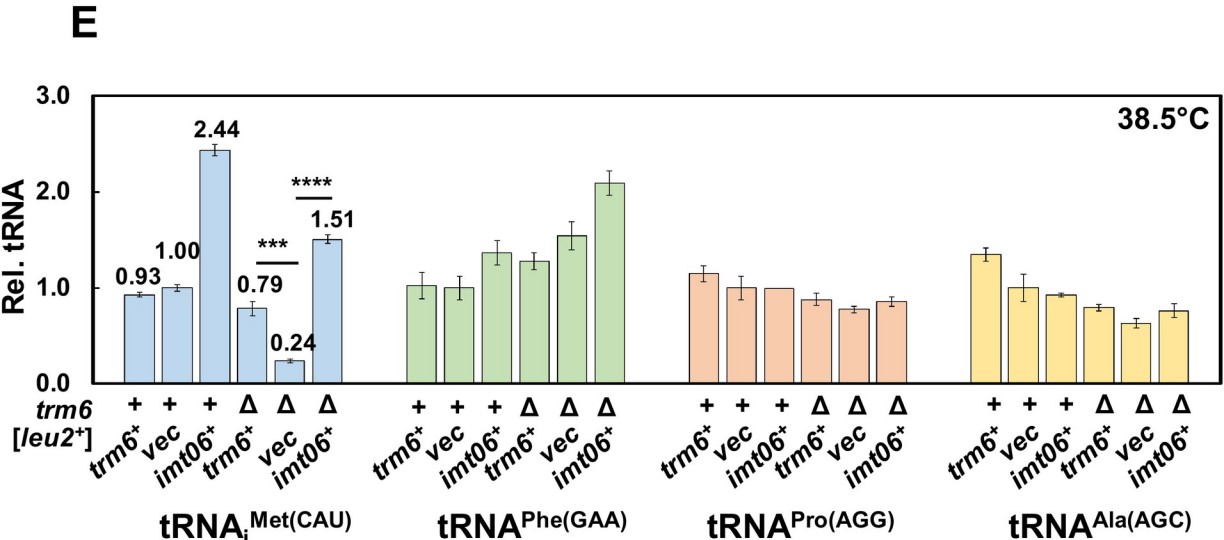

**Fig 2.** *S. pombe trm6Δ temperature sensitivity is associated with reduced tRNA$_i$$^{Met(CAU)}$ levels. (A) Northern analysis of tRNAs in S. pombe trm6Δ and WT cells before and after shift from 30°C to 38.5°C.* Strains were grown in YES media at 30°C, shifted to 38.5°C for 9 hours as

described in Materials and Methods, and RNA was isolated at the indicated times, and analyzed by northern blotting, with probes as indicated. tMi(CAU), tRNA$_i$$^{Met(CAU)}$; tF(GAA), tRNA$^{Phe(GAA)}$; tP(AGG), tRNA$^{Pro\ (AGG)}$; tA(AGC), tRNA$^{Ala(AGC)}$; tG(GCC), tRNA$^{Gly(GCC)}$. **(B) Quantification of tRNA levels in *S. pombe trm6Δ* and WT cells at 30˚C and 38.5˚C.** The bar chart depicts relative levels of tRNA species at each temperature, relative to their levels in the WT strain at 30˚C (each itself first normalized to levels of the control tG(GCC)). For each tRNA, the dark shade indicates 30˚C, and progressively lighter shades indicate time points (3, 6, 9 hours) Standard deviations for each tRNA measurement are indicated. The statistical significance of tRNA levels was evaluated using a two-tailed Student's t-test assuming equal variance. ns, not significant; *, $p < 0.05$; **, $p < 0.01$; ***, $p < 0.001$; ****, $p<0.0001$. tMi(CAU), blue; tF(GAA), green; tP(AGG), orange; tA(AGC), yellow. **(C) Overproduction of tRNA$_i$$^{Met(CAU)}$ suppresses the ts growth defects of *S. pombe trm6Δ* mutants.** Strains with plasmids as indicated were grown overnight in EMMC-Leu media at 30˚C and analyzed for growth as in Fig 1A on indicated plates and temperatures. **(D) Overproduction of tRNA$_i$$^{Met(CAU)}$ restores tRNA$_i$$^{Met(CAU)}$ levels in *S. pombe trm6Δ* mutants.** Strains containing plasmids as indicated were grown in EMMC-Leu at 30˚C and shifted to 38.5˚C for 8 hours as described in Materials and Methods, and RNA from cells grown at 38.5˚C was isolated and analyzed by northern blotting as in Fig 2A. **(E) Quantification of tRNA levels in *S. pombe trm6Δ* mutants and WT strains overproducing tRNA$_i$$^{Met(CAU)}$.** tRNA levels were quantified as in Fig 2B.

strain was completely suppressed by expression of the stand-alone *imt06⁺* gene (S5 Fig). As expected, northern analysis showed that *trm6Δ* strains expressing [*imt06⁺ leu2⁺*] had substantially more tRNA$_i$$^{Met(CAU)}$ than the *trm6Δ* [*leu2⁺*] vector control strains at 38.5˚C (6.3 fold; relative levels of 1.51 vs 0.24) while the levels of other tested tRNAs remained unchanged (Fig 2D and 2E). These results suggest strongly that the temperature sensitivity of *S. pombe trm6Δ* mutants is caused by the loss of tRNA$_i$$^{Met(CAU)}$ and that this tRNA is the major physiologically important tRNA substrate of Trm6:Trm61 methyltransferase.

## Mutations in the Rapid tRNA decay pathway restore growth and tRNA$_i$$^{Met(CAU)}$ levels in *S. pombe trm6Δ* mutants

To identify potential mechanisms that contribute to the loss of tRNA$_i$$^{Met(CAU)}$ in *S. pombe trm6Δ* mutants, we isolated and analyzed spontaneous suppressors of the temperature sensitivity. Among 25 temperature resistant suppressors from 14 cultures, we found three with increased levels of tRNA$_i$$^{Met(CAU)}$ but not of a control tRNA, and whole genome sequencing revealed that two of these had mutations in *dhp1⁺* (*dhp1-5* and *dhp1-6*) and one had a mutation in *tol1⁺* (*tol1-1*). *dhp1⁺* is the ortholog of *S. cerevisiae RAT1*, which encodes one of the two 5'-3' exonucleases involved in RTD, and *tol1⁺* is the ortholog of *S. cerevisiae MET22*, deletion of which inhibits RTD by inhibiting 5'-3' exonucleases [16–18].

Growth analysis on plates showed that the *trm6Δ dhp1-5* and *trm6Δ dhp1-6* mutants were nearly as healthy at high temperatures as the WT strain on both YES and EMMC-his media, whereas the *trm6Δ tol1-1* mutant was slightly less healthy at higher temperatures (Fig 3A). Northern analysis of tRNA from strains grown at 30˚C and after temperature shift to 38.5˚C showed that the *dhp1* and *tol1* suppressors substantially restored tRNA$_i$$^{Met(CAU)}$ levels at both high and low temperatures, without affecting any of a number of other tRNAs (Fig 3B and 3C). At 38.5˚C, tRNA$_i$$^{Met(CAU)}$ levels increased from 27% in *trm6Δ* strains (relative to WT at 30˚C) to 55%, 45%, and 32% in the *trm6Δ dhp1-5*, *trm6Δ dhp1-6*, and *trm6Δ tol1-1* mutants respectively, with no significant change in the levels of tRNA$^{Phe(GAA)}$, tRNA$^{Pro(AGG)}$, and tRNA$^{Ala(AGC)}$ (Fig 3B and 3C). This increase in tRNA$_i$$^{Met(CAU)}$ levels at 38.5˚C accounts for the temperature resistance of the strains, and reflects the weaker suppression in the *trm6Δ tol1-1* strain. At 30˚C, tRNA$_i$$^{Met(CAU)}$ levels also increased, from 53% of WT in the *trm6Δ* strains to 75%, 82%, and 82% in the *trm6Δ dhp1-5*, *trm6Δ dhp1-6*, and *trm6Δ tol1-1* mutants, again with little change in the levels of other tRNAs. Thus, it appears that the observed decay is occurring at both temperatures.

Two lines of evidence suggest that the *dhp1* mutations were responsible for the suppression in the *trm6Δ dhp1* mutants. First, the isolation of two different *dhp1* missense mutations in genetically independent suppressors argues strongly that the relevant suppressing mutation is

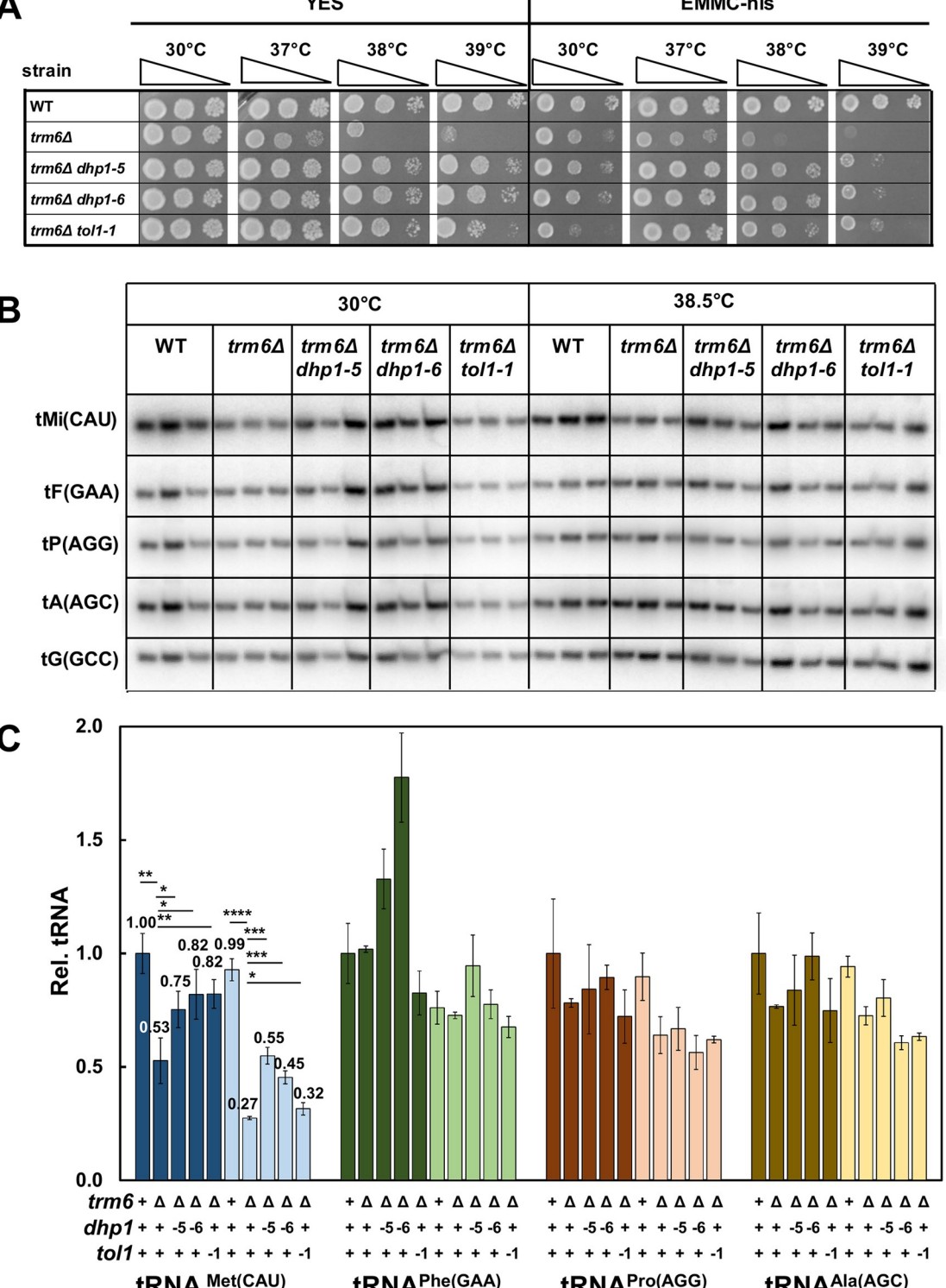

**Fig 3. Spontaneous suppressors of *S. pombe trm6Δ* mutants with mutations in *dhp1* and *tol1* restore growth and increase tRNAᵢ^Met(CAU) levels of *S. pombe trm6Δ* mutants.** *(A)* **Spontaneous suppressors of *S. pombe trm6Δ* mutants with mutations in *dhp1* and *tol1* restore growth at high temperatures.** Strains as indicated were grown overnight in YES media at 30°C and analyzed for growth as in Fig 1A on indicated plates and temperatures. *(B)* **Spontaneous suppressors of *S. pombe trm6Δ* mutants with mutations in *dhp1* and *tol1* restore tRNAᵢ^Met(CAU) levels after growth at 38.5°C.** Strains were grown in YES media at 30°C and

shifted to 38.5˚C for 8 hours as described in Materials and Methods, and RNA was isolated and analyzed by northern blotting as in Fig 2A. *(C)* **Quantification of tRNA levels of *S. pombe trm6Δ dhp1* and *tol1* mutants.** tRNA levels were quantified as in Fig 2B, with dark and light shades as indicated for 30˚C and 38.5˚C.

that in *dhp1*. Whole genome sequencing typically results in only a few mutations, and finding two independent suppressors with different *dhp1* mutations would be highly unlikely to occur by chance. Furthermore, like *S. cerevisiae RAT1*, *dhp1⁺* is an essential gene in *S. pombe*, and thus there are limited mutations that would reduce function without killing the cell. Indeed, alignments show that neither of the *dhp1* mutations is completely conserved, although each is likely to be important; the S737P (*dhp1-5*) mutation is predicted to disrupt the central portion of an α-helix, and the Y669C (*dhp1-6*) mutation is within a highly conserved block of amino acids (S6A and S6B Fig). Second, complementation experiments showed that introduction of an additional chromosomal copy of *dhp1⁺* at *ura4⁺* restored the temperature sensitivity of the *trm6Δ dhp1-5* mutant, to a level similar to that of a *trm6Δ* control strain. The increased gene dosage of *dhp1⁺* had no effect on growth of the WT strain, and a barely detectable inhibitory effect on growth of the *trm6Δ* strain (S6C Fig), which we attribute to the 2-fold overproduction of Dhp1 and the presumed sensitivity of the *trm6Δ* strain to any further reduction in tRNA$_i^{Met(CAU)}$ levels. We therefore conclude that the isolated *dhp1* mutations were responsible for the suppression of the temperature sensitive growth defect of *S. pombe trm6Δ* mutants.

Similarly, we infer that the *tol1-1* mutation is responsible for suppression because expression of P$_{tol1}$ *tol1⁺* on a [*leu2⁺*] plasmid restored temperature sensitivity to the *trm6Δ tol1-1* strain, with no effect on growth of the *trm6Δ* or the WT strain (S7 Fig). The *tol1-A151D* (*tol1-1*) point mutation is located in a highly conserved region of the essential *tol1⁺* gene [45], presumably resulting in a partial loss of function variant (S8A and S8B Fig).

The discovery of *dhp1* and *tol1* mutations as suppressors of the *S. pombe trm6Δ* temperature sensitivity demonstrates the involvement of the RTD pathway in decay of tRNA$_i^{Met(CAU)}$ lacking m$^1$A$_{58}$ in *S. pombe*. However, this result was unexpected, because it is well established in *S. cerevisiae* that *trm6* mutants trigger decay of pre-tRNA$_i^{Met(CAU)}$ by the nuclear surveillance pathway in vivo [13, 22, 24, 46] and in vitro [47].

## The exacerbated growth defect of an *S. pombe trm6Δ imt06Δ* mutant is due to further reduction in tRNA$_i^{Met(CAU)}$ levels, and suppressors of the growth defect are in the RTD pathway

To obtain a more robust set of suppressors of the *trm6Δ* temperature sensitivity, we further reduced tRNA$_i^{Met(CAU)}$ levels in the *trm6Δ* mutants by introduction of an *imt06Δ* mutation, decreasing the number of tRNA$_i^{Met(CAU)}$ genes from four to three. As anticipated, the resulting *trm6Δ imt06Δ* strain grew very poorly at 30˚C, and was temperature sensitive at higher temperatures, not growing at all at 37˚C, whereas the *imt06Δ* mutant had no growth defect at any tested temperature (Figs 4A and S9A). Moreover, the *trm6Δ imt06Δ* growth defect was strictly due to the loss of tRNA$_i^{Met(CAU)}$ because the *trm6Δ imt06Δ* strains expressing both an integrated copy of *imt06⁺* and a [*leu2⁺ imt06⁺*] plasmid grew as well as WT on YES media at all temperatures up to 39˚C (S9B Fig). As anticipated, tRNA$_i^{Met(CAU)}$ levels in the *trm6Δ imt06Δ* strain at 30˚C were substantially reduced from those in the *trm6Δ* mutants (40% vs 70% of WT), consistent with the 44% reduction in tRNA$_i^{Met(CAU)}$ levels in the *imt06Δ* strains (Figs 4B and S9C). As also expected, the levels of other tested tRNAs in these strains were virtually unaffected. These results show a prominent synthetic growth defect in the *S. pombe trm6Δ imt06Δ* strain, due only to reduced levels of tRNA$_i^{Met(CAU)}$.

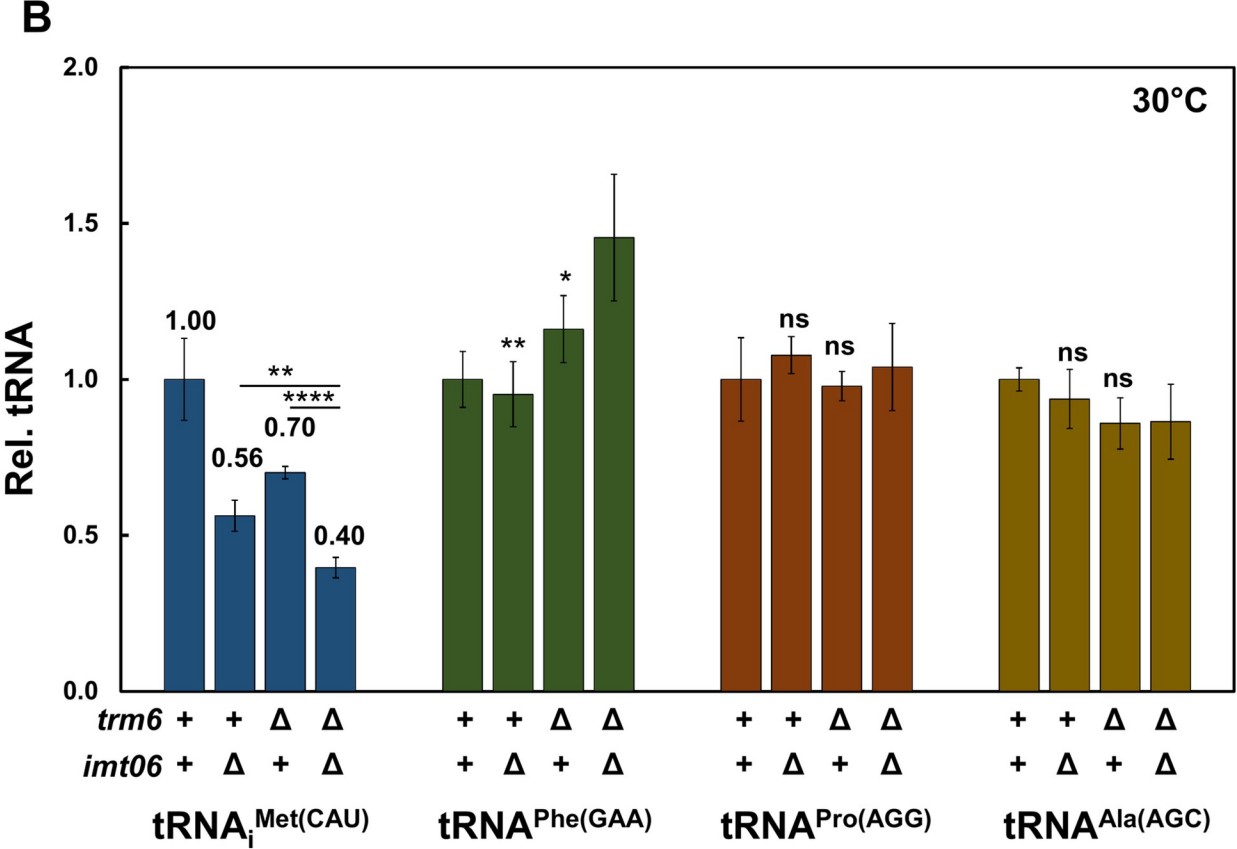

**Fig 4. Deletion of one of the four *S. pombe* genes encoding tRNAi^Met(CAU) in an *S. pombe trm6Δ* mutant exacerbates its growth defect and further reduces tRNAi^Met(CAU) levels.** *(A) Deletion of the imt06+ gene encoding tRNAi^Met(CAU) in an S. pombe trm6Δ mutant severely exacerbates its growth.* Strains were grown overnight in YES media at 30°C and analyzed for growth as in Fig 1A on indicated plates and temperatures for 4 days. *(B) Quantification of tRNAi^Met(CAU) levels in S. pombe trm6Δ imt06Δ mutants at 30°C.* tRNA levels were quantified as in Fig 2B.

Analysis of spontaneous suppressors of the severe growth defect of *trm6Δ imt06Δ* mutants revealed additional mutations in the RTD pathway. Of fifteen suppressors isolated from six independent cultures of *trm6Δ imt06Δ* strains after plating on YES media at 35°C, eleven had increased tRNA$_i$^Met(CAU) levels at both 30°C and 38.5°C, relative to a control tRNA, and whole genome sequencing of nine of these eleven suppressors revealed eight with *dhp1* mutations (six alleles; *dhp1-7* to *dhp1-12*), and one with a *tol1* mutation (*tol1-2*). All six new *dhp1* mutations and the *tol1-2* mutation were in conserved regions of the respective proteins (S10 Fig). Growth comparisons showed that the *dhp1-7*, *dhp1-8*, and *tol1-2* mutations each efficiently rescued the *trm6Δ imt06Δ* growth defect at 35°C, with the *trm6Δ imt06Δ tol1-2* strain growing almost as well as the original *trm6Δ* strain at 37°C (Fig 5A). Moreover, the growth phenotype of the *trm6Δ imt06Δ tol1-2* mutant was fully complemented upon introduction of a [*leu2*⁺ P$_{tol1}$ *tol1*⁺] plasmid (S11 Fig). Consistent with the growth suppression, tRNA$_i$^Met(CAU) levels were increased from 26% of WT in the *trm6Δ imt06Δ* mutant at 30°C to 49% and 55% in the corresponding *dhp1-7* and the *tol1-2* suppressors, and from 12% at 38.5°C to 35% and 34% in the suppressors, whereas control tRNA levels were largely unchanged at each temperature (Fig 5B and 5C).

These findings underscore the crucial role of Dhp1 and Tol1, and thus of the RTD pathway, in quality control of tRNA$_i$^Met(CAU) in *S. pombe trm6Δ* mutants. Indeed, the isolation of eight genetically distinct *dhp1/rat1* alleles (two in *trm6Δ* mutants and six in *trm6Δ imt06Δ* mutants) and two distinct *tol1/met22* alleles (one each in the *trm6Δ* mutant and the *trm6Δ imt06Δ* mutant) argues that the genetic landscape of *trm6Δ* suppressors has been nearly saturated, particularly considering that both *dhp1* and *tol1* are essential in *S. pombe*.

Further examination suggests the lack of participation of the Trf4 ortholog Cid14 of the nuclear surveillance pathway [48–50] in quality control of tRNA$_i$^Met(CAU) in *S. pombe trm6Δ* mutants. A *cid14Δ* mutation was introduced into WT and *trm6Δ* strains and independent isolates were confirmed by PCR (Materials and Methods), after which the resulting strains were shown to be sensitive to 5-fluorouracil (5-FU) (S12A and S12B Fig), as previously reported [48, 51]. We observed little, if any, suppression of the *trm6Δ* growth defect in the *trm6Δ cid14Δ* strains (S13A Fig), and only very minor restoration of tRNA$_i$^Met(CAU) levels at high temperature, relative to levels in *trm6Δ* mutants (21% vs 18%, compared to 39% in the *trm6Δ dhp1-5* strain) (S13B and S13C Fig). Thus, we infer that tRNA$_i$^Met(CAU) is degraded in *S. pombe trm6Δ* and *trm6Δ imt06Δ* mutants primarily by the RTD pathway, and not appreciably by the TRAMP complex of the nuclear surveillance pathway.

## A *met22Δ* mutation substantially suppresses the *S. cerevisiae trm6-504* temperature sensitivity and partially restores tRNA$_i$^Met(CAU) levels at low and high temperatures

Because of our discovery of the predominant role of the RTD pathway in tRNA$_i$^Met(CAU) quality control in *S. pombe trm6Δ* mutants, we examined the participation of the RTD pathway in tRNA$_i$^Met(CAU) quality control in the well-studied *S. cerevisiae trm6-504*^ts mutant, which had previously been shown to trigger tRNA$_i$^Met(CAU) decay by the nuclear surveillance pathway [13, 22, 23]. We found that deletion of *MET22* substantially suppressed the temperature sensitive growth defect of an *S. cerevisiae trm6-504*^ts mutant, both in its original background (Y190) and in the BY4741 (BY) background, with obvious suppression at 36°C in both backgrounds and at 37°C in the BY background (Fig 6A). Consistent with the growth suppression, tRNA$_i$^Met(CAU) levels were increased from 12% of WT in the BY *trm6-504* strain to 35% in the *met22Δ* derivative at 34°C, and from 38% to 54% at 27°C, with little effect on other tested tRNAs (Fig 6B and 6C). Similar restoration of tRNA$_i$^Met(CAU) levels was observed in the *met22Δ* derivative of the original

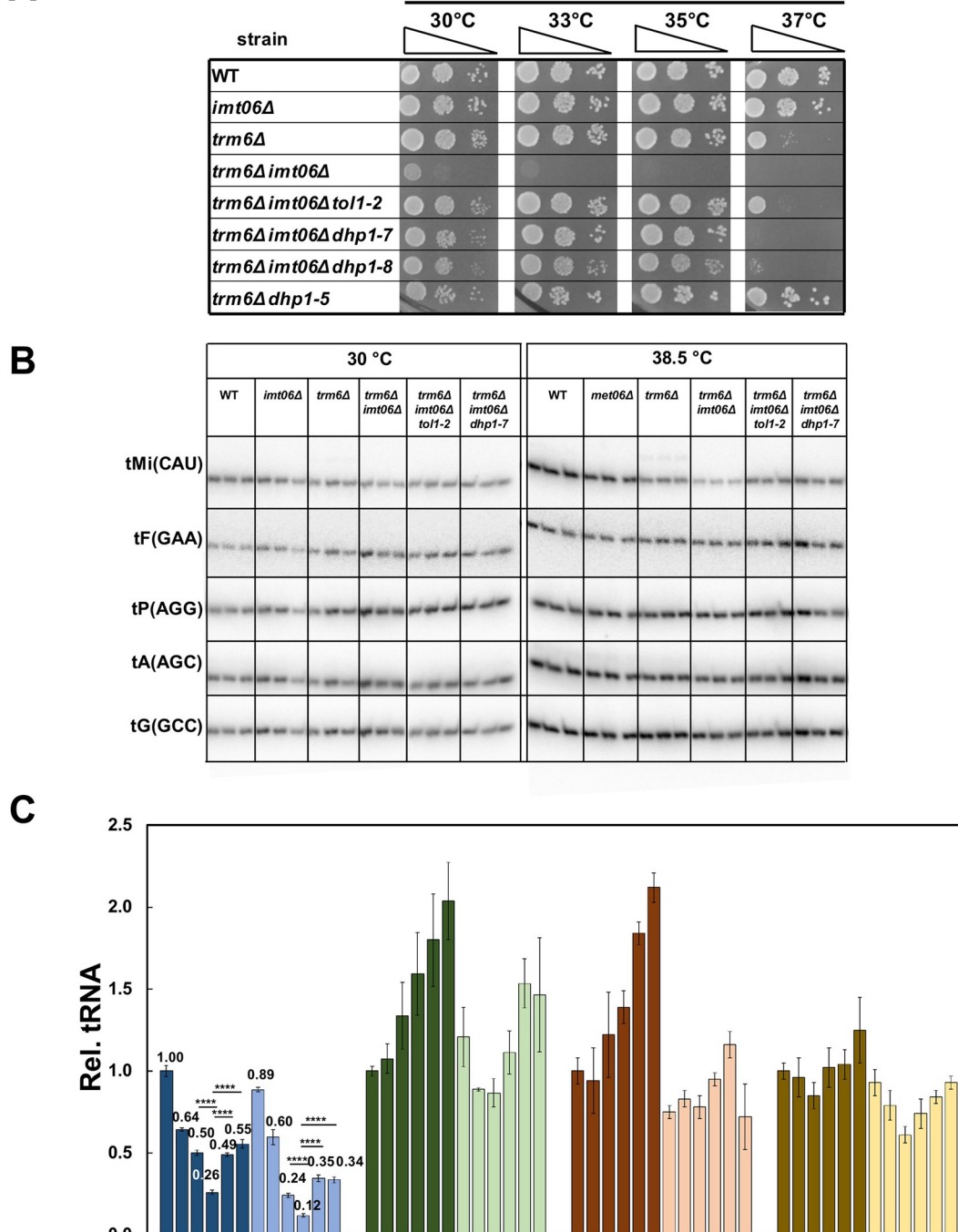

**Fig 5. Spontaneous suppressors of *S. pombe trm6Δ imt06Δ* mutants with mutations in *dhp1* and *tol1* restore growth and increase tRNA_i^Met(CAU) levels. *(A)* Spontaneous suppressors of *S. pombe trm6Δ imt06Δ* mutants with mutations in *dhp1* and *tol1* suppress the growth defect.** Strains were grown overnight in YES media at 30°C and analyzed for growth as in Fig 1A on indicated plates and temperatures. *(B)* **Spontaneous suppressors of *S. pombe trm6Δ imt06Δ* mutants with mutations in *dhp1* and *tol1* restore tRNA_i^Met(CAU) levels at low and high temperatures.** Strains were grown in YES media at 30°C and

shifted to 38.5°C for 6 hours as described in Materials and Methods, and RNA was isolated and analyzed by northern blotting as in Fig 2A. **(C) Quantification of tRNA$_i^{Met(CAU)}$ levels in *S. pombe trm6Δ imt06Δ dhp1-7* and *trm6Δ imt06Δ tol1-2* mutants.** tRNA levels were quantified as in Fig 2B.

Y190 *trm6-504* strain, with no effect on other tested tRNAs (S14 Fig). These results show that *MET22* regulates tRNA$_i^{Met(CAU)}$ levels in *trm6-504* strains regardless of their genetic background and suggest the involvement of the RTD pathway in tRNA$_i^{Met(CAU)}$ quality control in *S. cerevisiae trm6-504* mutants.

## Each of the RTD pathway exonucleases has a significant role in tRNA$_i^{Met(CAU)}$ quality control in *S. cerevisiae* BY *trm6-504* mutants

Although the restoration of growth and tRNA$_i^{Met(CAU)}$ levels in a BY *trm6-504 met22Δ* mutant suggested the involvement of the RTD pathway, we sought to provide additional evidence by directly testing the roles of the RTD pathway exonucleases Rat1 and Xrn1 in tRNA$_i^{Met(CAU)}$ quality control in the *trm6-504* mutant. As Rat1 is essential [52], we tested the role of Rat1 using the *rat1-107* mutation, which we had previously isolated as a suppressor of RTD in *trm8Δ trm4Δ* mutants [16].

We found that mutation of each of the RTD exonucleases efficiently suppressed both the temperature sensitivity and the reduced tRNA$_i^{Met(CAU)}$ levels of the BY *trm6-504* mutant. Whereas the *trm6-504* mutant was impaired for growth at 33°C and above, the *trm6-504 rat1-107* strain had healthy growth at 37°C and visible growth at 39°C, which was similar to that of the *trm6-504 met22Δ* strain, and the *trm6-504 xrn1Δ* strain grew up to 37°C (Fig 7A), despite the known growth defect of *xrn1Δ* mutants [16]. This growth suppression by mutation of each RTD component was nearly as efficient as that due to mutation of the nuclear surveillance components *RRP6* or *TRF4* (Fig 7A) [13]. Moreover, consistent with the suppression results, the temperature dependent decay of tRNA$_i^{Met(CAU)}$ in *trm6-504* mutants was efficiently suppressed by mutation of RTD components. Thus, after 6-hour temperature shift to 34°C, relative tRNA$_i^{Met(CAU)}$ levels in *trm6-504* mutants were restored from 16% to 32%, 67%, and 32% by *met22Δ*, *xrn1Δ*, and *rat1-107* mutations respectively, comparable to the 52% observed in a *trm6-504 trf4Δ* mutant (Fig 7B and 7C). Significant suppression of tRNA$_i^{Met(CAU)}$ levels was also found at 27°C. A parsimonious interpretation of these results is that all components of the RTD pathway are involved in tRNA$_i^{Met(CAU)}$ quality control in *trm6-504* mutants, and that the nuclear surveillance pathway and RTD pathway each contribute substantially to this tRNA$_i^{Met(CAU)}$ quality control.

## tRNA$_i^{Met(CAU)}$ in *S. cerevisiae trm6-504* mutants and suppressors is fully modified to m$^1$A$_{58}$ at both low and high temperatures

As *trm6-504* mutants are known to have reduced, but not absent, m$^1$A modification levels [13], we wanted to determine if m$^1$A levels were altered in tRNA$_i^{Met(CAU)}$ as a result of the temperature shift in *trm6-504* mutants. By using poison primer extension to measure m$^1$A$_{58}$ modification, we found that A$_{58}$ of tRNA$_i^{Met(CAU)}$ was nearly fully modified at both 27°C and 34°C in both *trm6-504* mutants (96% and 94%) and WT strains (98% and 97%) (S15 Fig), although tRNA$_i^{Met(CAU)}$ levels are reduced in *trm6-504* mutants. By contrast, A$_{58}$ of tRNA$^{Phe(GAA)}$ was substantially hypomodified at low temperature in *trm6-504* mutants compared to WT strains (25% vs 83%), and also at high temperature (27% vs 68%) (S15 Fig). As tRNA$_i^{Met(CAU)}$ is 96% modified at low temperature and present at 39% of WT levels, whereas tRNA$^{Phe(GAA)}$ is 25% modified and present at 97% of WT levels, these findings suggest that tRNA$_i^{Met(CAU)}$ is the

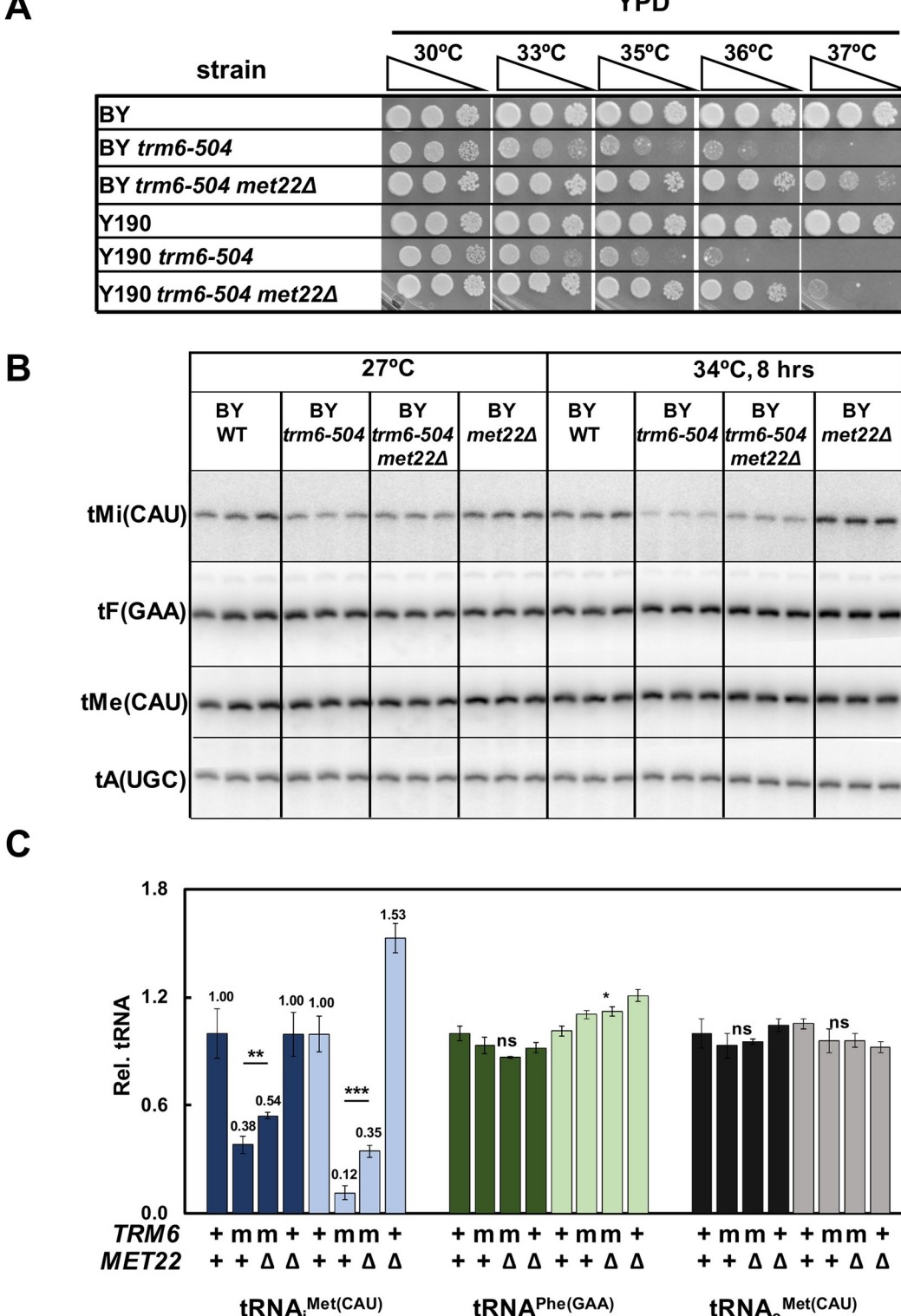

**Fig 6. The temperature sensitivity and reduced tRNAᵢ^Met(CAU) levels in *S. cerevisiae trm6-504* mutants are suppressed by a *met22Δ* mutation.** *(A) A met22Δ mutation substantially suppresses the S. cerevisiae trm6-504 temperature sensitivity.* Strains were grown overnight in YPD media at 30˚C and analyzed for growth on indicated plates and temperatures. BY; standard BY4741 WT strain background; Y190, background of original *trm6-504* mutant *(B) A met22Δ mutation substantially restores tRNAᵢ^Met(CAU) levels in S. cerevisiae trm6-504 mutants.* Strains were grown in YPD

media at 27˚C and shifted to 34˚C for 8 hours as described in Materials and Methods, and RNA was isolated and analyzed by northern blotting. *(C)* **Quantification of northern analysis of tRNA$_i^{Met(CAU)}$ levels in *S. cerevisiae* BY *trm6-504* mutants.** tRNA levels were quantified as in Fig 2B. m, *trm6-504* mutant.

preferred substrate of Trm6:Trm61. In addition, comparison of the tRNA$^{Phe(GAA)}$ modification levels in *trm6-504* mutants at low and high temperature suggests that there is little or no temperature-dependent reduction in the Trm6:Trm61 methyltransferase activity.

To further assess the connection between m$^1$A$_{58}$ modification and tRNA decay, we measured m$^1$A levels in tRNA$_i^{Met(CAU)}$ in *trm6-504* strains with mutations in the nuclear surveillance or the RTD pathway. We found that tRNA$_i^{Met(CAU)}$ was still nearly fully modified to m$^1$A$_{58}$ at 34˚C in *trm6-504* mutants with suppressing mutations in any of the components of the RTD pathway (*met22Δ*, *rat1-107* or *xrn1Δ*) or the nuclear surveillance pathway (*trf4Δ* or *rrp6Δ*), with modification levels ranging from 92.6% to 95.2%, compared to 97.6% in WT cells (S16A and S16C Fig). As anticipated, A$_{58}$ modification of tRNA$^{Phe(GAA)}$ was similarly reduced in *trm6-504* mutants and in derivatives with suppressing mutations in the RTD or nuclear surveillance pathway, compared to WT cells (S16B and S16C Fig). Thus, although tRNA$_i^{Met(CAU)}$ decay at 34˚C is inhibited in *trm6-504* strains with mutations in the nuclear surveillance or the RTD pathway, the nearly complete modification of the remaining, undegraded tRNA$_i^{Met(CAU)}$ in all of these *trm6-504* derivative strains argues for competition between the Trm6:Trm61 enzyme and the decay pathways.

## The lethality of *S. cerevisiae trm6Δ* mutants is suppressed by mutation of both the RTD and the nuclear surveillance pathways, but not either one alone

To separate the effects of the decay pathways from the presumed competition with Trm6: Trm61, we determined if the lethality of *S. cerevisiae trm6Δ* mutants could be rescued by inhibition of either or both of the RTD and nuclear surveillance pathways. As previously shown, the *S. cerevisiae trm6Δ* lethality is due to lack of tRNA$_i^{Met(CAU)}$ [21], as a *trm6Δ* [*TRM6 URA3*] [2μ *IMT1 LEU2*] strain was healthy when the *URA3* plasmid was selected against on media containing 5-FOA, but the corresponding strain with a [2μ *LEU2*] vector died (S17 Fig). We found that deletion of both *MET22* and *TRF4* suppressed the lethality of the *S. cerevisiae trm6Δ* mutant on 5-FOA-containing media at 30˚C and 33˚C, but neither single deletion could rescue the lethality of the *trm6Δ* mutant alone (Fig 8A and 8B). Thus, we conclude that both the RTD pathway and the nuclear surveillance pathway significantly contribute to tRNA$_i^{Met(CAU)}$ quality control in mutants lacking m$^1$A$_{58}$ modification. As tRNA$_i^{Met(CAU)}$ levels in *trm6Δ met22Δ trf4Δ* strains were only 48% of WT levels, comparable to the tRNA$_i^{Met(CAU)}$ levels in *trm6-504* mutants and somewhat less than tRNA$_i^{Met(CAU)}$ levels in mutants lacking one of the four *IMT* genes (Fig 8C and 8D), we infer that tRNA$_i^{Met(CAU)}$ lacking m$^1$A$_{58}$ modification is still being degraded in this strain.

## Discussion

We have provided strong evidence that the rapid tRNA decay pathway has a major role in decay of tRNA$_i^{Met(CAU)}$ lacking m$^1$A$_{58}$ in both *S. pombe* and *S. cerevisiae*. In *S. pombe*, *dhp1* and *tol1* mutations in the RTD pathway suppress the temperature sensitivity and decay of tRNA$_i^{Met(CAU)}$ in *trm6Δ* and *trm6Δ imt06Δ* mutants. In *S. cerevisiae*, *met22*, *rat1*, and *xrn1* mutations in the RTD pathway suppress the temperature sensitivity and tRNA$_i^{Met(CAU)}$ decay

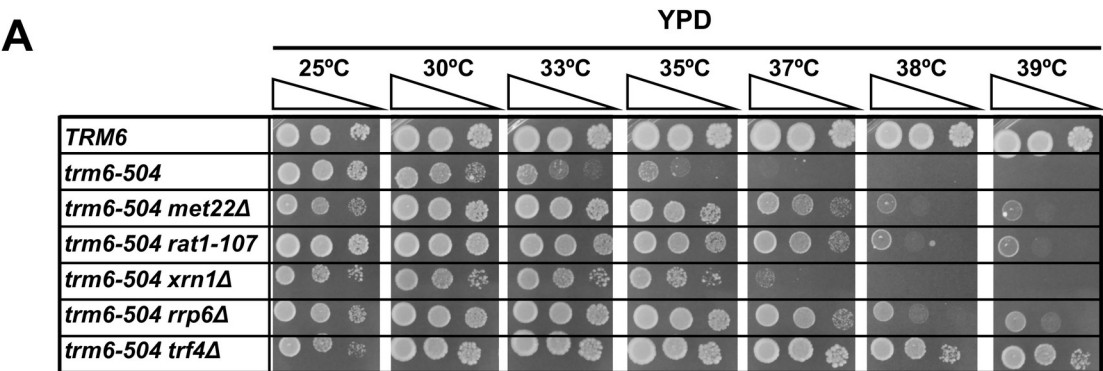

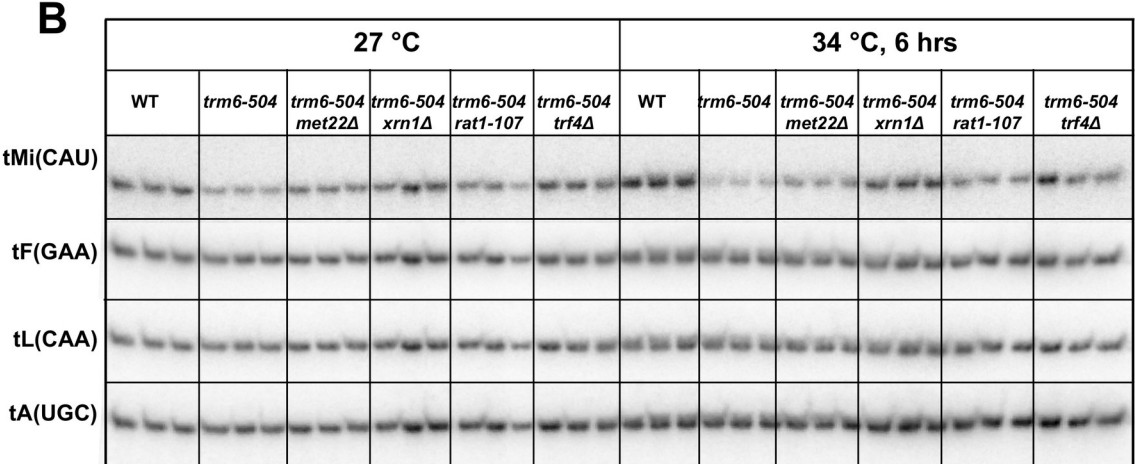

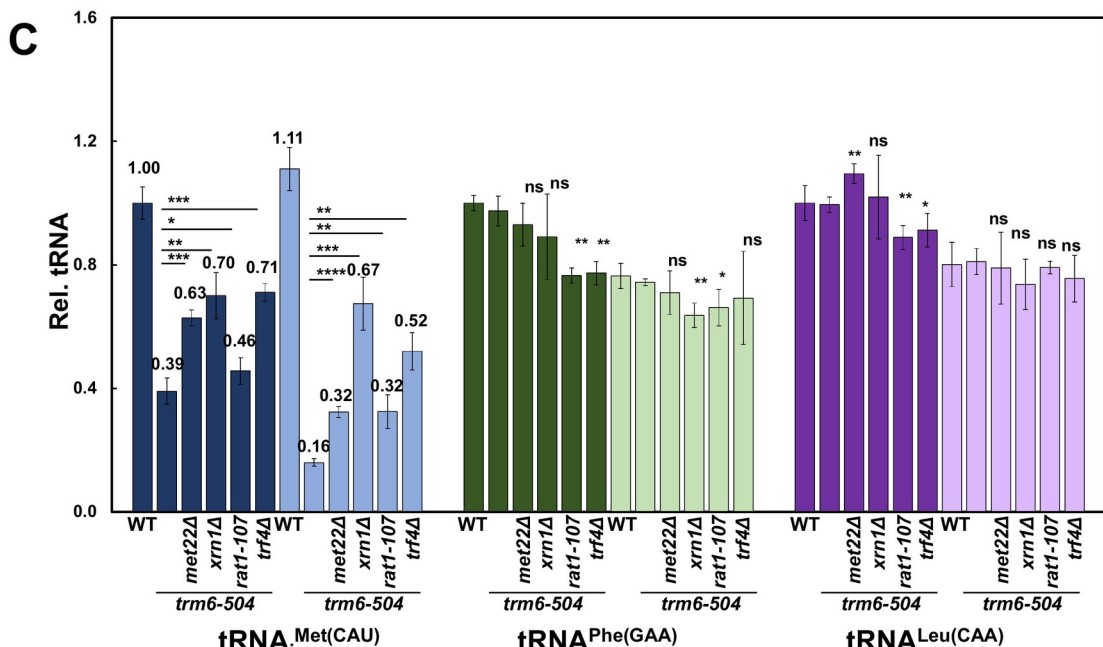

**Fig 7. The RTD and nuclear surveillance pathways are each involved in tRNAᵢ^Met(CAU) quality control in *S. cerevisiae trm6-504* mutants.** *(A)* **The *S. cerevisiae trm6-504* mutant growth defect is substantially suppressed by mutations in individual components of the RTD and nuclear surveillance pathways.** Strains were grown overnight in YPD media at 30°C and analyzed for growth on

indicated plates and temperatures. **(B) tRNA$_i$$^{Met(CAU)}$ levels are substantially restored in *S. cerevisiae trm6-504* strains with mutations in individual components of the RTD and nuclear surveillance pathways.** Strains were grown in YPD at 27˚C and shifted to 34˚C for 6 hours as described in Materials and Methods, and RNA was isolated and analyzed by northern blotting. **(C) Quantification of tRNA$_i$$^{Met(CAU)}$ levels in *S. cerevisiae trm6-504* strains with mutations in the RTD and nuclear surveillance pathways.** Dark and light colors indicate growth at 27˚C and 34˚C.

in *trm6-504* mutants, and a *met22Δ* mutation is required, together with a *trf4Δ* mutation in the nuclear surveillance pathway, to restore viability of *S. cerevisiae trm6Δ* mutants.

The finding that reduced tRNA$_i$$^{Met(CAU)}$ levels are the cause of the defect in both *S. pombe* and *S. cerevisiae trm6* mutants lacking m$^1$A$_{58}$ is consistent with the unique structural properties of eukaryotic tRNA$_i$$^{Met(CAU)}$ in the D-loop and the T-loop. In this region, eukaryotic initiator tRNA differs from that of canonical elongator tRNAs in the absence of N$_{17}$, the replacement of canonical residues with A$_{20}$, A$_{54}$, and A$_{60}$, and a unique substructure involving these residues and m$^1$A$_{58}$, as well as G$_{57}$ and A$_{59}$ [26, 53]. As Trm6:Trm61 are conserved in eukaryotes [38, 54], we infer that tRNA$_i$$^{Met(CAU)}$ levels will be similarly subject to decay in other eukaryotes lacking m$^1$A$_{58}$.

These findings establish that the RTD pathway acts on all body modification mutants that have been shown to result in decay of tRNAs in fungi, including *S. cerevisiae* mutants lacking m$^7$G$_{46}$, ac$^4$C$_{12}$, or m$^{2,2}$G$_{26}$, particularly in combination with other body modification mutants [16, 20], *S. pombe* mutants lacking m$^7$G$_{46}$ [15], and now mutants lacking m$^1$A$_{58}$ in both organisms. This set of results suggests that the RTD pathway will mediate decay of other body modification mutants in *S. pombe* and *S. cerevisiae*. Furthermore, as *S. cerevisiae* and *S. pombe* diverged about 600 Mya [39], these findings suggest that body modification mutants in other eukaryotes will also undergo decay by the RTD pathway. In support of this suggestion, we note that Rat1/Dhp1, Met22/Tol1, and Xrn1 of the RTD pathway are all conserved in eukaryotes [55, 56], and that WT HeLa cells (without a modification defect) that are incubated at 43˚C undergo decay of tRNA$_i$$^{Met(CAU)}$ by Rat1 and Xrn1 [35]. As a subset of hypomodified tRNAs are known to be reduced in mammalian cells lacking m$^7$G$_{46}$ [57, 58] or m$^5$C [59], it is likely that tRNA decay is occurring in these cells, and based on our results, we speculate that this decay is due to the RTD pathway.

It was surprising to find that the decay of tRNA$_i$$^{Met(CAU)}$ in *S. pombe trm6Δ* mutants was not due to the nuclear surveillance pathway, because of its well established role in decay of tRNA$_i$$^{Met(CAU)}$ in *S. cerevisiae trm6-504* mutants [13, 22, 23]. We argued above that in *S. pombe*, tRNA$_i$$^{Met(CAU)}$ lacking m$^1$A$_{58}$ is primarily degraded by the RTD pathway, because we had essentially saturated the genetic landscape of suppressors of *S. pombe trm6Δ* or *trm6Δ imt06Δ* strains with mutations in *dhp1* or *tol1* of the RTD pathway, and because a *cid14Δ* mutation in the nuclear surveillance pathway did not restore growth or tRNA$_i$$^{Met(CAU)}$ levels to an *S. pombe trm6Δ* mutant. It is known that the other components of the nuclear surveillance pathway are present and functional in *S. pombe* [48, 49, 60, 61]. It is possible that the lack of participation of the nuclear surveillance pathway in decay of tRNA$_i$$^{Met(CAU)}$ in *S. pombe trm6Δ* mutants is due in some way to the structure of three of the four tRNA$_i$$^{Met(CAU)}$ genes, each of which is present as a dimeric tRNA gene, and expressed as a tandem tRNA$^{Ser}$-tRNA$_i$$^{Met}$ transcript that is then processed into individual tRNAs [42]. Alternatively, it is possible that the lack of participation of the nuclear surveillance pathway in *S. pombe trm6Δ* mutants is due to some, as yet unappreciated, difference in the structure or folding between *S. pombe* and *S. cerevisiae* tRNA$_i$$^{Met(CAU)}$, or to differences in the activity of the nuclear surveillance pathway.

Our finding that *dhp1* and *tol1* mutations significantly restore tRNA$_i$$^{Met(CAU)}$ levels at 30˚C and 38.5˚C in *S. pombe trm6Δ* and *trm6Δ imt06Δ* mutants underscores that the RTD pathway

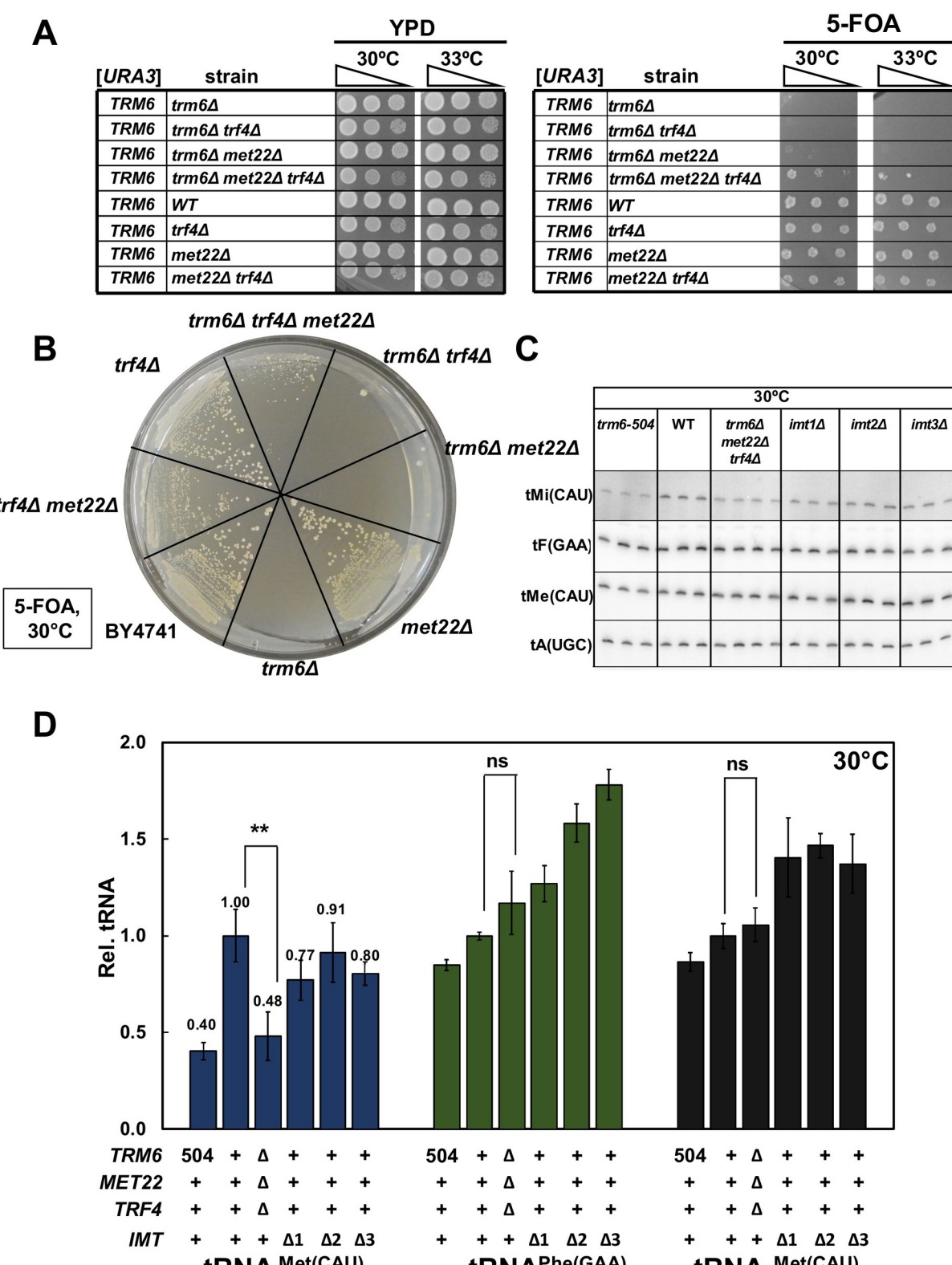

**Fig 8. The lethality of an *S. cerevisiae trm6Δ* mutant is suppressed by deletion of both *MET22* and *TRF4*, but not by either deletion alone, and results in modest tRNA$_i$^Met(CAU)^ levels.** *(A)* Growth test on plates of an *S. cerevisiae trm6Δ* [*URA3 TRM6*] strain with a *met22Δ* and/or *trf4Δ* mutation. Strains were grown overnight in YPD media at 30°C, diluted to OD$_{600}$~3, serially diluted 10-fold in water, and 2 μL were

spotted onto YPD or 5-FOA plates as indicated, and grown for 4 days. *(B)* **Pie streak growth test of an *S. cerevisiae* trm6Δ [*URA3 TRM6*] strain with a *met22Δ* and/or *trf4Δ* mutation.** Strains were grown overnight in YPD media at 30˚C, and then 2 μl of each of the cultures was streaked on 5-FOA plates for single colonies, and then incubated at 30˚C for 7 days. *(C)* **Northern analysis of tRNA$_i^{Met(CAU)}$ in *S. cerevisiae* trm6Δ met22Δ trf4Δ mutants.** Strains were grown in YPD at 30˚C for 6 hours as described in Materials and Methods, and RNA was isolated and analyzed by northern blotting. *(D)* **Quantification of tRNA$_i^{Met(CAU)}$ levels in *S. cerevisiae* trm6Δ met22Δ trf4Δ mutants.** tRNA levels were quantified as in Fig 2B.

is active at all temperatures in *S. pombe*, as we also found in *S. cerevisiae* trm6-504 mutants, and previously found in *S. cerevisiae* mutants lacking m$^7$G$_{46}$ and m$^5$C [16] and in fully modified variants of tRNA$^{Tyr}$ [62]. The relatively healthy growth of *S. pombe* trm6Δ mutants and the poor growth of trm6Δ imt06Δ mutants at 30˚C, with tRNA$_i^{Met(CAU)}$ levels at ~50% and 26% of WT respectively, is consistent with prior results that *S. cerevisiae* strains are healthy with three of four tRNA$_i^{Met(CAU)}$ genes, generally grow slowly with two of four genes, and can survive with as little as 40% of WT fully modified tRNA$_i^{Met(CAU)}$ [63, 64].

Our results provide evidence that the nuclear surveillance and the RTD pathways in *S. cerevisiae* are in competition with the Trm6:Trm61 m$^1$A methyltransferase, as tRNA$_i^{Met(CAU)}$ was fully modified in *S. cerevisiae* trm6-504 mutants at both 27˚C and 34˚C, and inhibition of either decay pathway resulted in more tRNA$_i^{Met(CAU)}$ that was fully modified. Moreover, as Trm6:Trm61 is a nuclear enzyme [21], and Xrn1 is cytoplasmic [65], the increased levels of fully modified tRNA$_i^{Met(CAU)}$ in trm6-504 xrn1Δ mutants argues that unmodified tRNA$_i^{Met(CAU)}$ is not immediately degraded, but rather that tRNA$_i^{Met(CAU)}$ goes to the cytoplasm without m$^1$A and returns to the nucleus for another chance at m$^1$A modification by Trm6:Trm61. This second chance for m$^1$A modification is analogous to the second chance pathway suggested earlier for tRNAs lacking m$^{2,2}$G$_{26}$ [66].

Our finding that tRNA$^{Phe(GAA)}$ in *S. cerevisiae* trm6-504 mutants had very similar m$^1$A$_{58}$ modification at both 27˚C and 34˚C suggests that Trm6:Trm61 is not temperature sensitive in this strain. If so, the reduced tRNA$_i^{Met(CAU)}$ levels at high temperature in *trm6-504* mutants would imply that tRNA$_i^{Met(CAU)}$ lacking m$^1$A$_{58}$ is itself temperature sensitive, perhaps becoming partially unfolded at high temperature. One could envision a model in which tRNA$_i^{Met(CAU)}$ lacking m$^1$A$_{58}$ is functioning in the cell cytoplasm (consistent with the viability of *S. pombe* trm6Δ mutants and of *S. cerevisiae* trm6Δ trf4Δ met22Δ mutants), but is in equilibrium with a state in which the tertiary structure is partially or completely unfolded due to lack of the modification [21, 26]. As tertiary structure unfolding precedes unfolding of the individual helices of tRNA [67], the disrupted tertiary structure due to lack of m$^1$A$_{58}$ could lead to unfolding of the acceptor stem and increased availability of the 5' end to the RTD pathway. This model is very similar to that we proposed previously to explain the increased Xrn1 susceptibility of *S. cerevisiae* tRNA$^{Ser(CGA)}$ lacking ac$^4$C$_{12}$ and Um$_{44}$ and for tRNA$^{Val(AAC)}$ lacking m$^7$G$_{46}$ and m$^5$C$_{49}$ [68], and could be tested in subsequent experiments.

## Materials and methods

### Yeast strains

All *S. pombe* and *S. cerevisiae* strains with integrated markers that are described in this work were made in biological triplicate. *S. pombe* strains are shown in S1 Table. *S. pombe* trm6Δ:: KanMX strains were constructed in the *S. pombe* WT strain derived from SP286 (*ade6-M210/ ade6-M216, leu1-32/leu1-32, ura4-D18/ura4-D18 h+/h+*) by PCR amplification of the *trm6Δ:: kanMX* cassette from the *S. pombe* trm6Δ::kanMX strain of the genomic knockout collection [40], followed by linear transformation using lithium acetate [69], and PCR screening of transformants for the presence of the deletion. Other *S. pombe* deletion strains were made similarly,

but the DNA containing the drug marker (*KanMX* or *HygR*) was obtained by Gibson assembly of ~ 500 nt 5' of the target site, the drug marker, and ~ 500 nt 3' of the target site [70]. The *dhp1*+ and *imt06*+ genes were integrated at the chromosomal *ura4-D18* locus using a single *ura4*+integrating vector containing the corresponding gene under its promoter [71]. *S. cerevisiae* deletion strains are shown in S2 Table, and were constructed by linear transformation with PCR amplified DNA from the appropriate knockout strain, followed by PCR amplification to confirm the knockout.

The *S. cerevisiae trm6-504* mutant strain was obtained in two ways. We obtained the original *trm6-504* (Y200) and its WT parent (Y190) from Dr. James Anderson. The BY *trm6-504* strain was reconstructed essentially as previously described [72], with three DNA components constructed in a plasmid vector: first, nt 893–1434 of the *TRM6* coding sequence (containing the C1292G mutation of the *trm6-504* mutant) and 204 nt of the 3' UTR; second, *K. lactis URA5*; third, nt 1384–1434 of the *TRM6* coding region. The DNA construct was removed from the vector, transformed into *S. cerevisiae* WT cells by linear transformation, confirmed by PCR, and then strains were plated onto media containing 5-FOA to select for Ura⁻ cells obtained by homologous recombination, which were sequence verified.

## Plasmids

Plasmids used in this study are listed in S3 Table. The *S. pombe* plasmid expressing *S. pombe* $P_{trm6}$ *trm6*+, *S. pombe* $P_{trm61}$ *trm61*+, and $P_{tol1+}$ *tol1*+ contained ~ 1000 bp and 500 bp of flanking 5' and 3' DNA and were cloned into a pREP3X-derived plasmid, removing the $P_{nmt1}$ promoter, as described [15]. Plasmids expressing *S. pombe* tRNA genes contained ~ 300 bp of flanking 5' and 3' DNA. The *S. pombe ura4*+ single integrating vectors [71] expressing *imt06*+ or $P_{dhp1+}$ *dhp1*+ were constructed similarly.

## Yeast media and growth conditions

*S. pombe* strains were grown in rich (YES) media or Edinburgh minimal complete (EMMC) media, or corresponding dropout media, as described [15]. For temperature shift experiments, cells were grown in YES or EMMC-leu media at 30°C to $OD_{600}$ ~ 0.5, and then diluted to $OD_{600}$ 0.1 in pre-warmed media at the desired temperature, and grown to $OD_{600}$ ~ 0.5, and then aliquots were chilled, harvested at 4°C, washed with ice cold water, frozen on dry ice, and stored at -80°C. *S. cerevisiae* strains were grown in rich (YPD) media or minimal complete (SDC) media as described [15], and temperature shift experiments were performed as described for *S. pombe*. All experiments with measurements were performed in biological triplicate.

## Spontaneous suppressor isolation

Spontaneous suppressors of *S. pombe trm6Δ* and *trm6Δ imt06Δ* mutants were isolated by growing cultures of individual colonies in YES media at 30°C, followed by plating 10⁷ cells on YES and EMMC plates at 39°C (for *trm6Δ* mutants) and on YES plates at 35°C (for *trm6Δ imt06Δ* mutants).

## Bulk RNA preparation and northern blot analysis

For northern analysis, biological triplicates were grown in parallel, aliquots of 3–5 OD were harvested, and then bulk RNA was prepared with glass beads and phenol as described [73], resolved on a 10% polyacrylamide (19:1), 7M urea, 1X TBE gel, transferred to Amersham Hybond-N+ membrane (Cytiva, Marlborough, MA cat# RPN303B), and hybridized with 5'

[32]P-labeled DNA probes (S4 Table) as described [14], followed by exposure and imaging on an Amersham Typhoon phosphorimager (Cytiva, Marlborough, MA), and quantification using Image Quant v5.2

## Isolation and purification of bulk tRNA

*S. pombe* WT and *trm6Δ* mutant strains were grown to ~ 0.5 OD in YES media at 30°C, and then bulk low molecular weight RNA was extracted from ~ 300 OD of pellets by using hot phenol [74], and resolved on an 8% polyacrylamide (19:1) 7M urea, 1X TBE gel to purify bulk tRNA by elution of tRNA from the excised gel slice.

## Isolation and purification of tRNA$^{Tyr(GUA)}$

tRNA$^{Tyr(GUA)}$ was purified from *S. pombe* WT strains, and *trm6Δ* and *trm61Δ* mutant strains using 1 mg of bulk RNA (prepared using hot phenol), and the 5'-biotinylated oligonucleotide (TDZ 365; tY(GUA) 76–64; 5' TGGTCTCCTGAGCCAGAATCGAACTA 3'), as described [74].

## HPLC analysis of nucleosides

Purified tRNA$^{Tyr(GUA)}$ (~ 1.25 µg) was digested to nucleosides by treatment with P1 nuclease, followed by phosphatase, as described [74], and nucleosides were analyzed by HPLC (Waters, Millford, MA) at pH 7.0 as described [75]. To quantify nucleosides in bulk tRNA, relative amounts of modified nucleosides were compared to cytidine.

## Poison primer extension assays

Oligomers used for primer extension are shown in S5 Table. Primers were 5'-end labeled essentially as described [74], with excess label removed using a MicroSpin G-25 chromatography column (Cytiva, Marlborough, MA cat#27532501), and poison primer extension was done as described [76], in 10 µL reactions containing 2 U AMV reverse transcriptase (Promega, Madison, WI cat# M5101), 1X AMV RT buffer, 1 mM ddNTP, and 1 mM of the other three dNTPs. Following extension for 1 h at 50°C, aliquots were resolved on a 15% polyacrylamide gel (29:1) containing 7 M urea in 1× TBE, and the gel was dried on a Model 583 Biorad gel dryer, exposed and analyzed on an Amersham Typhoon phosphorimager, and quantified using Image Quant v5.2.

## Whole genome sequencing

Whole genome sequencing was performed by the University of Rochester Genomics Center at 25–50 fold coverage of the genome, and reads were compared to the corresponding parent strain, and to the reference genome.

## Supporting information

**S1 Fig. The temperature sensitivity of *S. pombe trm6Δ* and *trm61Δ* mutants is complemented by expression of the corresponding gene.** *(A) The temperature sensitivity of an S. pombe trm6Δ mutant is complemented by [P$_{trm6}$ trm6$^+$ leu2$^+$] on EMMC-leu media.* WT and *trm6Δ* cells expressing P$_{trm6}$ *trm6+* were grown overnight in EMMC-leu media 30°C and analyzed for growth at the indicated temperatures. *(B) The temperature sensitivity of S. pombe trm61Δ is complemented by [P$_{trm61}$ trm61$^+$ leu2$^+$] on EMMC-leu media.* WT and *trm61Δ* cells expressing P$_{trm61}$ *trm61+* were grown overnight in EMMC-leu media 30°C and

analyzed for growth.
(PDF)

**S2 Fig. *S. pombe trm6Δ* mutants lack m$^1$A in their bulk tRNA. *(A,B)* Bulk tRNA from *S. pombe trm6Δ* mutants have no detectable m$^1$A.** *S. pombe trm6Δ* mutants and WT cells were grown in biological triplicate in YES media at 30°C and bulk tRNA was purified, digested to nucleosides, and analyzed for modifications by HPLC as described in Materials and Methods. *(A) A trace of the A$^{258 \text{ nm}}$ of eluted nucleosides of bulk tRNA. (B) Quantification of levels of modified nucleosides of purified tRNA$^{Tyr(GUA)}$*. The bar chart depicts the average moles/mol of nucleosides (expressed as a percentage of the moles of cytidine), with associated standard deviation; WT, gray; *S. pombe trm6Δ*, red. The data is also tabulated in the table below *(C)*.
(PDF)

**S3 Fig. Northern analysis of all tested tRNAs in *S. pombe trm6Δ* and WT cells after shift from 30°C to 38.5°C. (A) Northern blot.** Full analysis is shown of tRNAs analyzed in the northern blot shown in Fig 2A. *(B,C)* **Quantification of tRNA levels.** The bar chart depicts relative levels of tRNA species at each temperature, relative to their levels in WT at 30°C.
(PDF)

**S4 Fig. Northern analysis of tRNA$_i$$^{Met(CAU)}$ levels in WT and *trm6Δ* strains expressing *imt06$^+$* or *sup9$^+$-imt07$^+$* from a [*LEU2*] plasmid. (A) Northern blot.** Strains were grown and analyzed as in Fig 2D. **(B) Quantification of tRNA levels.** tRNA levels were quantified as in Fig 2B.
(PDF)

**S5 Fig. Overproduction of tRNA$_i$$^{Met(CAU)}$ suppresses the temperature sensitive growth defects of *S. pombe trm61Δ* mutants.** Strains with plasmids as indicated were grown overnight in EMMC-Leu media at 30°C and analyzed for growth as in Fig 1A on indicated plates and temperatures.
(PDF)

**S6 Fig. The *dhp1-5* and *dhp1-6* mutations that suppress the *S. pombe trm6Δ* growth defect disrupt conserved regions or structures of Dhp1. *(A)* Alignment of regions around the *dhp1-5 (S737P)* and *dhp1-6 (Y669C)* mutations.** *S. pombe* Dhp1 was aligned with putative Rat1/Dhp1 orthologs from 12 evolutionarily distinct eukaryotes, using Multalin [77]; http://multalin.toulouse.inra.fr/multalin/). red, more than 80% conservation; blue 40% - 80% conservation. *(B)* **Location of *dhp1-5 (S737P)* and *dhp1-6 (Y669C)* mutations mapped onto the *S. pombe* structure [78].** magenta, residues in the catalytic center; blue, residues interacting with Rai1. *(C)* **Expression of P$_{dhp1}$ *dhp1+* integrated in the chromosome restores temperature sensitive growth of the *S. pombe trm6Δ dhp1-5* mutant.** WT, *trm6Δ*, and *trm6Δ dhp1-5* cells expressing a chromosomally integrated copy of P$_{dhp1}$ *dhp1+* in the *ura4+* locus or the control vector integrant, were grown overnight in YES media at 30°C, and analyzed for growth.
(PDF)

**S7 Fig. The suppression of the *S. pombe trm6Δ* growth defect by a *tol1-1* mutation is complemented by expression of *tol1* on a [P$_{tol1}$ *tol1$^+$ leu2$^+$*] plasmid.** *trm6Δ tol1-1* mutants, *trm6Δ* mutants and WT strains were transformed with either [P$_{tol1}$ *tol1$^+$ leu2$^+$*] or empty vector, grown in EMMC-leu, and spotted.
(PDF)

**S8 Fig. The isolated *tol1-1* mutation that restores growth of an *S. pombe trm6Δ* mutant is in a conserved region of the protein.** *(A)* **Alignment of the regions around the *tol1-1* (*A151D*) mutation.** *S. pombe* Tol1 was aligned with putative Tol1 orthologs from 12 evolutionarily distinct eukaryotes, as in S5 Fig. red, more than 80% conservation; blue 40% - 80% conservation. *(B)* **Location of *tol1-1* (*A151D*) mapped onto the structure of the *S. cerevisiae* ortholog Met22** [79]. orange, active site residues.
(PDF)

**S9 Fig. Deletion of one of the four *S. pombe* genes encoding tRNA$_i^{Met(CAU)}$ in an *S. pombe trm6Δ* mutant exacerbates its growth defect and further reduces tRNA$_i^{Met(CAU)}$ levels.** *(A)* **Deletion of the *imt06* gene encoding tRNA$_i^{Met(CAU)}$ in an *S. pombe trm6Δ* mutant severely exacerbates its growth.** Strains from the growth test in Fig 4A are shown after 2 days of growth *(B)* **Complementation of *trm6Δ imt06Δ* growth defect with an integrated *imt06* and a [*leu2$^+$ imt06$^+$*] plasmid.** *trm6Δ imt06Δ* and WT cells expressing tRNA$_i^{Met(CAU)}$ from a chromosomally integrated copy of *imt06+* and from a [*leu2$^+$ imt06+*] plasmid, and controls were grown overnight in EMMC or EMMC-leu media at 30˚C, and analyzed for growth. *(C)* **Levels of tRNA$_i^{Met(CAU)}$ are significantly reduced in *S. pombe trm6Δ imt06Δ* mutants at 30˚C.** The Northern blot from Fig 4B is shown.
(PDF)

**S10 Fig. Suppressor mutations isolated in *S. pombe trm6Δ imt06Δ* strains are in conserved regions of Dhp1 and Tol1.** *(A)* **Alignment of regions around the *dhp1* suppressor mutations.** The alignment of *S. pombe* Dhp1 was done as in S5A Fig. *(B)* **Location of *dhp1* suppressor mutations mapped onto the *S. pombe* structure** [78]. magenta, residues in the catalytic center; blue, residues interacting with Rai1. *(C)* **Alignment of the regions around the *tol1-2* (*A297D*) mutation.** The alignment of *S. pombe* Tol1 was done as in S7 Fig. *(D)* **Location of *tol1-2* (*A297D*) mapped onto the structure of the *S. cerevisiae* ortholog Met22** [79]. orange, active site residues.
(PDF)

**S11 Fig. Expression of [P$_{tol1}$ tol1$^+$ leu2$^+$] fully complements the *S. pombe trm6Δ imt06Δ tol1-2* mutants.** WT, *trm6Δ imt06Δ*, and *trm6Δ imt06Δ tol1-2* cells expressing P$_{tol1}$ tol1+ or a vector [80] were grown overnight in EMMC-Leu media at 30˚C, and analyzed for growth
(PDF)

**S12 Fig. A *cid14Δ* mutation causes 5-FU sensitivity in *S. pombe* WT and *trm6Δ* strains.** *(A)* **Analysis of growth of *cid14Δ* strains on YES media with or without 5-FU.** Strains were grown overnight in YES media at 30˚C and analyzed for growth as in Fig 1A on indicated plates and temperatures. *(B)* **Complementation of the 5-FU sensitivity of *cid14Δ* strains.** Strains were grown overnight in EMMC-Leu media at 30˚C and analyzed for growth on indicated plates and temperatures.
(PDF)

**S13 Fig. A *cid14Δ* mutation does not suppress the growth defect of *S. pombe trm6Δ* mutants and has only a minimal effect on tRNA$_i^{Met(CAU)}$ levels.** *(A)* **A *cid14Δ* mutation does not suppress the growth defect of *S. pombe trm6Δ* mutants.** Strains were grown overnight in YES media at 30˚C and analyzed for growth as in Fig 1A on indicated plates and temperatures. *(B,C)* **A *cid14Δ* mutation has only a minimal effect on tRNA$_i^{Met(CAU)}$ levels in *S. pombe trm6Δ* mutants.**
(PDF)

**S14 Fig. A *met22Δ* mutation partially restores tRNA$_i$$^{Met(CAU)}$ levels in *S. cerevisiae* Y190 *trm6-504* mutants.** Strains were grown in YPD at 27˚C and shifted to 33˚C for 6 hours as described in Materials and Methods, and RNA was isolated and analyzed by northern blotting. *(A)* **Northern Blot.** *(B)* **Quantification of northern.** B; standard BY4741 WT strain background; Y, Y190 background of original *trm6-504* mutant; m, *trm6-504* mutant.
(PDF)

**S15 Fig. In *S. cerevisiae trm6-504* mutants, tRNA$_i$$^{Met(CAU)}$ is fully modified to m$^1$A$_{58}$ at 27˚C and 34˚C, while tRNA$^{Phe(GAA)}$ is hypomodified to a similar extent at both temperatures.** *(A)* **Primer extension analysis of m$^1$A$_{58}$ modification in tRNA$_i$$^{Met(CAU)}$ and tRNA$^{Phe(GAA)}$**. Bulk RNA from *S. cerevisiae trm6-504* mutants and WT cells grown for Fig 7B was analyzed by poison primer extension assay, as described in Materials and Methods, with the P1 primer (complementary to tRNA$_i$$^{Met(CAU)}$ nt 76–61) and P2 primer (complementary to tRNA$^{Phe(GAA)}$ 76–60) in the presence of ddCTP, producing a stop at G$_{57}$ for both tRNA$_i$$^{Met(CAU)}$ and tRNA$^{Phe(GAA)}$, and a stop at N$_{59}$ for m$^1$A$_{58}$. *(B)* **Quantification of the poison primer extension.** Values were calculated by first subtracting background levels, as in Fig 1E.
(PDF)

**S16 Fig. In *S. cerevisiae trm6-504* mutants grown at 34˚C, tRNA$_i$$^{Met(CAU)}$ is fully modified to m$^1$A$_{58}$ in derivatives with mutations in the RTD and nuclear surveillance pathways.** *(A)* **Primer extension analysis of m$^1$A$_{58}$ modification in tRNA$_i$$^{Met(CAU)}$**. Bulk RNA from the growth done for Fig 7B was analyzed by poison primer extension assay with the P1 primer in the presence of ddATP, producing a stop at U$_{55}$ or at A$_{59}$ for m$^1$A$_{58}$. *(B)* **Primer extension analysis of m$^1$A$_{58}$ modification in tRNA$^{Phe(GAA)}$**. Bulk RNA from the growth done for Fig 7B was analyzed by poison primer extension assay with the P2 primer in the presence of ddCTP, producing a stop at G$_{57}$ and for m$^1$A$_{58}$ at U$_{59}$. *(C)* **Quantification of the data from *(A)* and *(B)*.**
(PDF)

**S17 Fig. Overexpression of *S. cerevisiae IMT1* fully suppresses *trm6Δ* mutant lethality.** *S. cerevisiae* WT, *trm6-504*, and *trm6Δ* strains containing [*2μ* P$_{GAL}$*TRM6 URA3*] plasmid [81] and [*2μ IMT1 LEU2*] plasmids or empty vector, as indicated, were grown overnight in SD-leu media at 30˚C and analyzed by spotting on SD-Leu media containing 5-FOA. Then cells from the 5-FOA plates were streaked for colonies, inoculated into SD-Leu media and grown overnight, and re-spotted on SD-Leu media.
(PDF)

**S1 Table. *S. pombe* strains used in this study.**
(PDF)

**S2 Table. *S. cerevisiae* strains used in this study.**
(PDF)

**S3 Table. Plasmids used in this study.**
(PDF)

**S4 Table. Oligomers used for northern analysis.**
(PDF)

**S5 Table. Oligomers used for primer extension analysis.**
(PDF)

## Acknowledgments

We thank Jeffrey Pleiss for *S. pombe* strains and James Anderson for the *S. cerevisiae trm6-504* and background strain. We also thank Elizabeth Grayhack, Thareendra de Zoysa, Alayna Hauke, Erin Marcus, and other members of the Phizicky and Grayhack labs for valuable discussions and comments during the course of this work, and Elizabeth Grayhack for critical reading of the manuscript.

## Author Contributions

**Conceptualization:** Monika Tasak, Eric M. Phizicky.

**Data curation:** Monika Tasak.

**Formal analysis:** Monika Tasak, Eric M. Phizicky.

**Funding acquisition:** Eric M. Phizicky.

**Investigation:** Monika Tasak.

**Methodology:** Monika Tasak, Eric M. Phizicky.

**Project administration:** Eric M. Phizicky.

**Resources:** Eric M. Phizicky.

**Supervision:** Eric M. Phizicky.

**Validation:** Monika Tasak.

**Writing – original draft:** Monika Tasak.

**Writing – review & editing:** Monika Tasak, Eric M. Phizicky.

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
