## [Decision Letter · Decision Letter 0]

23 May 2022

Dear Dr Phizicky,

Thank you very much for submitting your Research Article entitled 'Initiator tRNA lacking 1-methyladenosine is unexpectedly targeted by the rapid tRNA decay pathway in evolutionarily distant yeast species' to PLOS Genetics.

The manuscript was fully evaluated at the editorial level and by independent expert peer reviewers. As you will see, the reviewers appreciated the importance of the work and were enthusiastic about publication but identified some valid constructive concerns that we ask you to address in a revised manuscript.  One reviewer noted that the claim that decay of tRNAiMet(CAU) in S. pombe trm6Δ mutants does not involve the nuclear surveillance pathway is not fully justified by the results in Fig. S11 and suggested measuring tRNAiMet(CAU) levels in the appropriate mutants and controls.

We therefore ask you to modify the manuscript according to the review recommendations. Your revisions should address the specific points made by each reviewer.

[LINK]

Yours sincerely,

Richard J. Maraia, M.D.

Guest Editor

PLOS Genetics

Gregory P. Copenhaver

Editor-in-Chief

PLOS Genetics

Reviewer's Responses to Questions

**Comments to the Authors:**

Reviewer #1: This paper provides strong evidence that the rapid tRNA decay (RTD) pathway has a major role in decay of initiator tRNA (tRNAiMet(CAU)) lacking m1A58 methylation in both S. pombe and S. cerevisiae, appearing to be the sole pathway in S. pombe, and functioning independently of the nuclear surveillance pathway in S. cerevisiae. It was not known previously that the RTD pathway functions on the tRNAiMet(CAU) lacking m1A58 in S. cerevisiae, in which the instability of this hypomodified tRNAiMet(CAU) has been well established. These findings are significant in showing that RTD pathway acts on all tRNA body modification mutations that evoke decay in fungi; and the evolutionary conservation of the RTD enzymes in mammalian cells suggests its involvement in controlling tRNAiMet(CAU) degradation in mammals, which has medical importance.

In general, the work was extremely well done, the quality of the data very high, and the presentation of results and interpretations very clear. There are however a few issues to be addressed.

Major comments:

-line 280-281: shouldn’t the trm6Δ imt06Δ mutant with the integrated imt06+ gene show poor growth at 39C compared to WT cells (the phenotype of trm6∆?)

-lines 417-419: The claim that decay of tRNAiMet(CAU) in S. pombe trm6Δ mutants does not involve the nuclear surveillance pathway is not fully justified by the results in Fig. S11. They need to measure tRNAiMet(CAU) levels in the trm6∆ and trm6∆cid14∆ mutants examined for growth in Fig. S11 and show no detectable increase in tRNA level in the trm6∆cid14∆ double mutant vs trm6∆ mutant, including a trm6∆dhp1-7 double mutant as a positive control. Also, can they verify the successful deletion of CID14 in their double mutant by showing a growth or biochemical phenotype of cid14∆ that can be complemented by WT CID14?

--lines 303-306: no mutations in the MET22 ortholog were recovered. Doesn’t this argue against saturation ot trm6∆ suppressors in S. pombe? Does deletion of the MET22 ortholog suppress the trm6∆ growth defect in S. pombe?

-the finding in Fig. S13A-B, that A58 of tRNAiMet(CAU) is nearly fully modified at both 27°C and 34°C in trm6-504 mutants is initially surprising, considering the genetic and biochemical evidence for reduced stability of tRNAiMet(CAU) in this strain, shown here and previously.

They state that “The nearly complete modification of tRNAiMet(CAU) in all of the trm6-504 strains with mutations in the nuclear surveillance or the RTD pathway argues for competition between the Trm6:Trm61 enzyme and the decay pathways.” It would be helpful to elaborate on this conclusion by reminding the reader in this context that the levels of tRNAiMet(CAU) are greatly reduced by trm6-504, to <20% of WT levels, and that the remaining ~20% is fully modified. This would be expected if trm6-504 strongly impairs, but does not abolish, m1A modification of tRNAiMet(CAU) and the unmodified species are essentially eliminated from the cells by degradation. Otherwise, there is an impression from the data in Fig. S13A-B that trm6-504 has no impact on tRNAiMet(CAU) methylation.

-There are unexplained data in Fig. 8C-D involving deletions of IMT1, 2, 3 genes. Also, the data for the trm6-504 mutant in this experiment were not mentioned in the text.

Other comments:

-p. 9: Is the failure of high-copy imt07+ to suppress the trm6/61 mutants due to a lack of overexpression of tRNAiMet(CAU, as judged by Northern?

-the methods used for isolating the spontaneous suppressors of trm6∆ should be described.

-Fig. 5B: typo: met06∆ should be imt06∆

Reviewer #2: Post-transcriptional modifications of tRNAs play critical roles in tRNA stability and translation. Previous studies established that the m1A58 modification, which occurs on multiple tRNAs, is critical for the stability of the initiator methionyl-tRNA (tRNA(iMet)). The genes encoding the enzymes responsible for the m1A58 modification, TRM6/GCD10 and TRM61/GCD14, are essential in S cerevisiae but can be deleted in S pombe. Whereas previous studies in S cerevisiae demonstrated that tRNA(iMet) turnover occurred through the nuclear surveillance pathway, a key finding of the present study is that turnover of tRNAs lacking the m1A58 modification in S pombe occurs through the cytoplasmic rapid tRNA decay (RTD) pathway. The authors first characterized the roles of TRM6 and TRM61 in S pombe and discovered that loss of these genes causes a temperature-sensitive growth defect due to reduced levels of tRNA(iMet). Characterization of spontaneous fast-growing suppressors of a trm6 mutant revealed mutations in dhp1, encoding a 5’-3’ exonuclease in the RTD pathway, and in tol1, encoding a phosphatase/nucleotidase that degrades a byproduct that feedback inhibits the RTD pathway. These mutations stabilized tRNAs in the absence of the m1A58 modification, indicating that tRNAs lacking the modification are typically degraded by the RTD pathway. Next, the authors asked if the RTD pathway also contributed to turnover of tRNA(iMet) in S cerevisiae trm6 mutants. Consistent with this possibility, the authors found that mutation of MET22, the tol1 ortholog in S cerevisiae, as well as mutations in the RTD exonucleases RAT1 and XRN1, suppressed the growth defect and loss of tRNA(iMet) in a trm6 mutant. Finally, the authors show that simultaneous inhibition of both the nuclear surveillance pathway (trf4 mutation) and the RTD pathway (met22 mutation) is sufficient to suppress the lethal phenotype of a trm6 deletion mutant in S cerevisiae. Thus, the authors conclude that both the nuclear surveillance pathway and the RTD pathway function in S cerevisiae to turnover tRNAs lacking the m1A58 modification.

This is an impressive and well-controlled study. The paper is well-written and the data support the authors’ conclusions.

Specific comments:

1. I wonder if it would be good to mention to readers the original names for TRM6 (GCD10) and TRM61 (GCD14) in yeast. Perhaps the field has fully adapted to the TRM nomenclature; however, some readers might still search for papers using the old GCD names.

2. Lines 270-1: editorial issue, should “(13, 22) (24, 46)” be combined to a single citation for all four papers?

3. Line 309: is there a positive control that could be used to confirm the cid14-deletion mutation? While PCR assays were used to confirm the mutations, it would be reassuring to see an expected phenotypic effect of the cid14-deletion. However, I am not aware if such a test exists.

4. Lines 347-348: At this point in the paper, is it possible that the RTD mutants might be indirectly affecting tRNA(iMet) levels through the nuclear surveillance pathway? While later data rule this out, it seems possible that RTD mutants could impair the nuclear surveillance pathway (perhaps impair expression of an enzyme in the pathway) and thereby affect tRNA(iMet) levels. Might want to word soften the conclusion here.

5. Lines 359-360: based on the observation that all tRNA(iMet) has the m1A58 modification, the authors posit that tRNA(iMet) is a preferred substrate of Trm6:Trm61. However, an alternative hypothesis is that tRNA(iMet) lacking the m1A58 modification is turned over more rapidly than other tRNAs lacking this modification. Thus, it is not more rapid modification, but more rapid turnover of unmodified tRNA that is responsible for high levels of modified tRNA(iMet) in the mutant cells.

6. Lines 386-387: while the results in the trm6-deletion mutant demonstrate that both the RTD and nuclear surveillance pathways contribute to turnover of hypomodified tRNA(iMet), I am wondering if the two pathways are fully separate or if there might be some overlap. In Figure 7C it is shown that tRNA(iMet) levels are at 0.16 in the trm6-504 mutant and increase to 0.32 in the met22 mutant, 0.67 in the xrn1 mutant, and 0.52 in the trf4 mutant. I am wondering what happens in the trm6-504 background when both pathways are inactivated (met22 trf4 or xrn1 trf4 double mutants). Are the effects of the mutants additive or perhaps less than additive because the two pathways, even though physically separated, can sometimes act on the same tRNAs?

Reviewer #3: The manuscript by Tasak and Phizickyand titled “Initiator tRNA lacking 1-methyladenosine is unexpectedly targeted by the rapid tRNA decay pathway in evolutionarily distant yeast species” provides important new information to the field of tRNA biology. There work provides insights into the regulation of tRNA, translation initiation and the competition between modification and degradation, with the yeast conservation informing human tRNA biology. They have used elegant genetic, molecular and biochemical studies highlighting he importance of writers and modifications on tRNA stability. They have also used well developed mutant strains and whole genome sequencing for suppressor mutations as a powerful tool, and to provide finer details on tRNA degradation pathways and new details on a key substrate for the RTD pathway. This was a well-planned and executed study with an extensive set of logical and well controlled experiments. Their data rich manuscript has 8 regular and 15 supplementary figure panels, which highlights the rigor of their work. I am enthusiastic about the manuscript and believe that the findings that both the RTD and nuclear surveillance pathways contribute to tRNAiMet(CAU) quality control warrants publication in PloS Genetics. I suggest that the manuscript be accepted pending minor revisions (below).

Line 455 – the authors should include a figure for their model.

Line 546 - Details on how whole genome sequencing was done at UofR and should be included in the Materials and methods.

As m1A is regulated by Alkbh in humans and these are not found in budding yeast (not sure about fission yeast), the authors could speculate on these differences. Is that a potential rational for why S pombe do not use the nuclear surveillance pathway in decay of tRNAiMet(CAU)?

Minor comments.

I suggest removing “unexpected” from the title.

I suggest the authors omit or refine this sentence from the abstract “In contrast studies have shown that S. cerevisiae trm6 mutants with reduced 1-methyladenosine (m1A58) specifically target pre-tRNAiMet(CAU) to the nuclear surveillance pathway for 3’-5’ exonucleolytic decay 35 by the TRAMP complex and nuclear exosome. “ Keep it simple in the abstract and address the re-examined findings in the introduction.

In materials and methods please include city, state for all company names and catalog numbers for specific products.

Line 164: move “fig” to before the numbers. Similar for lines 264, 266, 293 and 297. Check throughout.

Line 168 – indicate how the tRNA modification was examined.

Line 180 – delete “all” as there may be some very minor residual m1A (0.03%)

Line 209 – Define for clarity – stand-alone tRNA met gene? - SPBTRNAMET.06 (imt06+) gene

**Have all data underlying the figures and results presented in the manuscript been provided?**

Reviewer #1: Yes

Reviewer #2: Yes

Reviewer #3: Yes

PLOS authors have the option to publish the peer review history of their article (what does this mean?). If published, this will include your full peer review and any attached files.

Reviewer #1: No

Reviewer #2: No

Reviewer #3: No

---

## [Decision Letter · Decision Letter 1]

5 Jul 2022

Dear Dr Phizicky,

We are pleased to inform you that your manuscript entitled "Initiator tRNA lacking 1-methyladenosine is targeted by the rapid tRNA decay pathway in evolutionarily distant yeast species" has been editorially accepted for publication in PLOS Genetics. Congratulations!

Yours sincerely,

Richard J. Maraia, M.D.

Guest Editor

PLOS Genetics

Gregory P. Copenhaver

Editor-in-Chief

PLOS Genetics

Comments from the reviewers (if applicable):

Reviewer's Responses to Questions

**Comments to the Authors:**

Reviewer #1: I am fully satisfied with the thorough responses to all of my questions and criticisms, which were completely resolved.

Reviewer #2: The authors have satisfactorily addressed all of my concerns.

Reviewer #3: The authors have addressed my concerns.

**Have all data underlying the figures and results presented in the manuscript been provided?**

Reviewer #1: Yes

Reviewer #2: Yes

Reviewer #3: None

PLOS authors have the option to publish the peer review history of their article (what does this mean?). If published, this will include your full peer review and any attached files.

Reviewer #1: No

Reviewer #2: No

Reviewer #3: No

**Data Deposition**

http://datadryad.org/submit?journalID=pgenetics&manu=PGENETICS-D-22-00494R1

**Press Queries**

---

## [Editor Report · Acceptance letter]

22 Jul 2022

PGENETICS-D-22-00494R1 

Initiator tRNA lacking 1-methyladenosine is targeted by the rapid tRNA decay pathway in evolutionarily distant yeast species 

Dear Dr Phizicky, 

We are pleased to inform you that your manuscript entitled "Initiator tRNA lacking 1-methyladenosine is targeted by the rapid tRNA decay pathway in evolutionarily distant yeast species" has been formally accepted for publication in PLOS Genetics! Your manuscript is now with our production department and you will be notified of the publication date in due course.

With kind regards,

Zsofia Freund

PLOS Genetics

On behalf of:
